

# Contributions of Catchment and In-Stream Processes to Suspended Sediment Transport in a Dominantly Groundwater-Fed Catchment

Yan Liu[1], Christiane Zarfl[1], Nandita B. Basu[2], Marc Schwientek[1], Olaf A. Cirpka[1]

[1]Center for Applied Geoscience, University of Tübingen, Tübingen, 72074, Germany

[2]Department of Civil and Environmental Engineering, University of Waterloo, Waterloo, ON N2L 3G1, Canada

*Correspondence to*: Olaf A. Cirpka (olaf.cirpka@uni-tuebingen.de)

**Abstract.** Suspended sediments impact stream water quality by increasing the turbidity and acting as a vector for strongly sorbing pollutants. Understanding their sources is of great importance to develop appropriate river management strategies. In this study, we present an integrated sediment transport model composed of a catchment-scale hydrological model to predict

river discharge, a river-hydraulics model to obtain shear stresses in the channel, a sediment-generating model, and a river sediment-transport model. We use this framework to investigate the sediment contributions from catchment and in-stream processes in the Ammer catchment close to Tübingen in South-West Germany. The model is calibrated to stream flow and suspended-sediment concentrations. We use the monthly mean suspended-sediment load to analyze seasonal variations of different processes. The contributions of catchment and in-stream processes to the total loads are demonstrated by model

simulations under different flow conditions. The evaluation of shear stresses by the river-hydraulics model allows identifying hotspots and hot moments of bed erosion for the main stem of the Ammer River. The results suggest that the contributions of suspended-sediment loads from urban areas and in-stream processes are higher in the summer months, while deposition has small variations with a slight increase in summer months. The catchment input, bed erosion, and bank erosion increase with an increase in flow rates. Bed erosion and bank erosion are negligible when flow is smaller than the corresponding

thresholds of 1.5 $m^3$ $s^{-1}$ and 2.5 $m^3$ $s^{-1}$, respectively. The bed-erosion rate is higher during the summer months and varies along the main stem. Over the simulated time period, net sediment trapping is observed in the Ammer River. The present work is the basis to study particle-facilitated transport of pollutants in the system, helping to understand fate and transport of sediments and sediment-bound pollutants.

## 1 Introduction

Suspended sediments are comprised of fine particulate matter (Bilotta and Brazier, 2008), which is an important component of the aquatic environment (Grabowski et al., 2011). Sediment transport plays significant roles in geomorphology, e.g., floodplain formation (Kaase and Kupfer, 2016), and transport of nutrients, such as particulate phosphorus and nitrogen (Haygarth et al., 2006;Slaets et al., 2014;Scanlon et al., 2004). Fine sediments are important for creating habitats for aquatic organisms (Amalfitano et al., 2017;Zhang et al., 2016). Conversely, high suspended sediment concentrations can have



negative impacts on water quality, especially, by facilitating transport of sediment-associated contaminants, such as heavy metals (Mukherjee, 2014;Peraza-Castro et al., 2016;Quinton and Catt, 2007) and hydrophobic organic pollutants such as polycyclic aromatic hydrocarbons (PAHs) (Rügner et al., 2014;Schwientek et al., 2013;Dong et al., 2015;Dong et al., 2016), polychlorinated biphenyls (PCBs), and other persistent organic pollutants (Meyer and Wania, 2008;Quesada et al., 2014).

Without understanding the transport of particulate matter, stream transport of strongly sorbing pollutants cannot be understood.

An efficient approach to estimate suspended-sediment loads is by rating curves, relating concentrations of suspended sediments to discharge. By this empirical approach, however, we cannot gain any information on the sources of suspended sediments, which is important for the assessment of particle-bound pollutants. Therefore, a model considering the various

processes leading to the transport of suspended sediments in streams is needed. Numerous sediment-transport models have been developed during the past decades, including empirical and physically based models. Commonly used empirical models include the Universal Soil Loss Equation (USLE) (Wischmeier and Smith, 1978) and the Sediment Delivery Distributed (SEDD) model (Ferro and Porto, 2000). The USLE was designed to estimate the long-term average annual rate of erosion on a field slope. The SEDD model considers morphological effects at annual and event scales. The two models are capable to

estimate sediment production from hillslopes, but cannot estimate sediment generation by in-stream processes. Among the models simulating physical processes, the Water Erosion Prediction Project (WEPP) (Flanagan and Nearing, 1995), the EUROpean Soil Erosion Model (EUROSEM) (Morgan et al., 1998), the Soil and Water Assessment Tool (SWAT) (Neitsch et al., 2011), the Storm Water Management Model (SWMM) (Rossman and Huber, 2016), the Hydrological Simulation Program Fortran (HSPF) model (Bicknell et al., 2001), and the Hydrologic Engineering Center's River Analysis System

(HEC–RAS) (Brunner, 2016) are widely used. WEPP and EUROSEM are applied to simulate soil erosion from hillslopes on the timescale of single storm events. The two models don't have the capability of estimating urban particles. SWAT uses a modified USLE method to calculate soil erosion from catchments. SWMM aims at simulating runoff quantity and quality from primarily urban areas, including particle accumulation and wash-off in urban areas. HSPF considers pervious and impervious land surfaces. None of these models represent in-stream processes well, especially not in natural river channels.

Various sediment-transport models for river channels exist that rely on detailed river hydraulics, particularly the bottom shear stress, which controls the onset of erosion and the transport capacity of a stream for a given grain diameter (Zhang and Yu, 2017;Siddiqui and Robert, 2010). HEC–RAS can be used to obtain detailed information on river hydraulics.

In this study, we present a numerical modeling framework to understand the combined contributions from catchment and in-stream processes to suspended-sediment transport. The main objectives of this study were: (i) to develop an integrated

sediment-transport model taking sediment-generating processes (e.g., particle accumulation and particle wash-off), and river sediment-transport processes (e.g., bed erosion and bank erosion) into consideration; (ii) to understand annual load and seasonal variations of suspended sediments from different processes; (iii) to investigate how the contributions of suspended sediments from catchment and in-stream processes change under different flow conditions; and (iv) to identify hotspots and hot moments of bed erosion.





## 2 Model Setup

### 2.1 Model Structure

The integrated sediment-transport model consists of a catchment-scale hydrological model, a river-hydraulics model, a catchment sediment-generating model and a river sediment-transport model (Fig. 1). The catchment-scale hydrological

5  model is used to estimate river discharge along the entire stream. The river-hydraulics model uses the discharge of the hydrological model and the river bathymetry to compute the river stage, cross-sectional area, velocity, and bottom shear stress, which are needed for the river-transport model. Towards this ends, we use HEC–RAS in quasi steady-state mode. The catchment sediment-generating model is used for simulating particle accumulation in urban areas during dry weather periods, particle wash-off during storms, and erosion from non-urban areas during rain periods. The river sediment-transport

10  model is used to simulate in-stream processes (advection, dispersion, deposition, as well as bank and bed erosion). Wastewater treatment plants (WWTPs) are treated as point inputs with constant discharge and sediment concentration during dry weather periods. Under low-flow conditions, when no soil erosion and urban particle wash-off occur and the suspended sediment concentrations in the streams are relatively small, we use a constant concentration to represent the sediment input under these conditions.

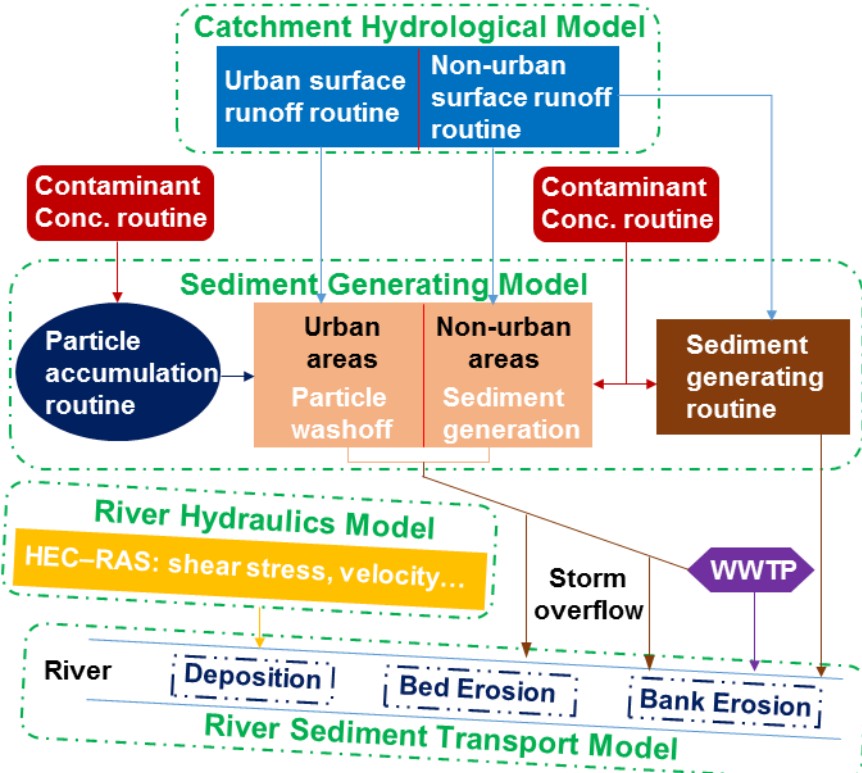

**Figure 1:** Integrated sediment transport model, consisting of a catchment-scale hydrological model, a river-hydraulic model, a sediment-generating model, and a river sediment-transport model.



## 2.2 Catchment-Scale Hydrological Model

The catchment-scale hydrological model is based on the HBV model (Hydrologiska Byråns Vattenbalansavdelning), by adding a quick recharge component and an urban surface runoff component to explain the special behavior of discharge in the Ammer catchment (see Sect. 3.1), which is the catchment where we apply the model. The main Ammer springs are fed

by groundwater from the karstified middle-Triassic Muschelkalk formation. The measured hydrograph indicates a rapid increase of base flow in sporadic events. We explain this behavior with a model that contains three storages of water in the subsurface: soil moisture in the top soils, a subsurface storage in the deeper unsaturated zone, and groundwater in the karstic aquifer. When the storage in the subsurface layer reaches a threshold, quick groundwater recharge occurs, which causes a rapid increase of base flow. Details of the hydrological model are given in the Appendix A. The temporal resolution of the

hydrological model is one hour. We use the catchment-scale hydrological model to simulate discharge contributions from 14 sub-catchments (detailed information see Sect. 3.1).

## 2.3 River-Hydraulics Model

In order to better understand in-stream processes, we feed the discharge data of the hydrological model into the river-hydraulics model HEC–RAS (Brunner, 2016), which solves the one-dimensional St.-Venant equations. The HEC–RAS

simulates hourly quasi-steady flow. The hourly discharge of the 14 sub-catchments simulated by the hydrological model is the input of HEC–RAS as change of discharge. The locations where the discharge from 14 sub-catchments enters into the main channel are set to the corresponding cross sections. Then HEC–RAS model computes the hourly hydraulics for the all cross-sections of the main channel and two major tributaries of the Ammer River. The distances between computed cross sections range from 10 m to 100 m depending on the changes of river bathymetry. The model requires river profiles in

regular cross-sections and yields detailed information for each cross section along the river channel, such as the water-filled cross-sectional area, the water depth, flow velocity, and shear stress, which are needed in the river sediment-transport model.

## 2.4 Sediment-Generating Model

The land use is classified into urban areas and non-urban areas. Impervious surfaces such as roads and roofs are regarded as urban areas, while non-urban areas consist of pervious surfaces such as gardens and parks, agricultural areas, and forests.

The sediment generating processes are different for the two types of land use. The sediment-generating model is used to obtain hourly sediments of different sources from the 14 sub-catchments.

### 2.4.1 Urban Areas

We use the urban-area algorithm of SWMM, which performs well on particle buildup and wash-off for urban land use, to describe sediment generation from urban areas. The corresponding processes are described below.

**(1) Particle Accumulation**





An exponential function is used to simulate particle accumulation during dry periods under the assumption that particles in the urban areas have a capacity, which is governed by the accumulation process during dry periods.

$$\frac{dM}{dt} = kM_{max}e^{-kt} \tag{1}$$

in which $M(t)$ [g m$^{-2}$] and $M_{max}$ [g m$^{-2}$] represent the particle buildup at a given time and the maximum buildup, respectively; $k$ [s$^{-1}$] is the rate constant for particle accumulation, and $t$ [s] denotes current time. The maximum buildup depends on the particle production and cleaning frequency, which is obtained through calibration.

**(2) Particle Wash-Off**

A power function is used to simulate particle wash-off during rain periods. The particle wash-off quantity is a function of surface runoff and the initial buildup of the corresponding rain period.

$$\frac{dM}{dt} = r_w = -k_w q^{n_w} M \tag{2}$$

$$c_{sw} = -\frac{r_w}{q} \tag{3}$$

in which $r_w$ [g m$^{-2}$ s$^{-1}$], $q$ [m s$^{-1}$], and $c_{sw}$ [mg L$^{-1}$] are the rate of wash-off, the surface runoff velocity, and the concentration of washed suspended sediment, respectively; $k_w$ [s$^{n_w-1}$ m$^{-n_w}$] and $n_w$ [-] represent a wash-off coefficient and a wash-off exponent.

**2.4.2 Non-Urban Areas**

In contrast to urban areas, the supply of suspended sediments from non-urban areas can be seen as "infinite" because they mainly originate from eroded soils. Soil erosion is assumed to linearly depend on shear stress, provided that the shear stress generated by surface runoff is larger than a critical shear stress. The sediment generation from non-urban areas is based on the study of Patil et al. (2012).

$$\tau = \rho_w g R_{surface} \sin\theta \tag{4}$$

$$y_h = \begin{cases} C_h(\tau - \tau_c) & if \ \tau > \tau_c \\ 0 & otherwise \end{cases} \tag{5}$$

$$c_{sed} = \frac{y_h}{q} \tag{6}$$

in which $\tau$ [N m$^{-2}$] is the mean shear stress generated by the average depth of surface runoff $R_{surface}$ [m], $\sin\theta$ [-] is the mean slope of the sub-catchment, $\rho_w$ [kg m$^{-3}$] is the density of water, and $g$ [m s$^{-2}$] is the gravitational acceleration constant. The non-urban sediment load $y_h$ [kg m$^{-2}$ s$^{-1}$] is directly proportional to the difference between the mean shear stress $\tau$ and the critical non-urban shear stress $\tau_c$ [N m$^{-2}$]. $C_h$ [s m$^{-1}$] is a proportionality constant. $c_{sed}$ [kg m$^{-3}$] is the concentration of sediment generated in non-urban areas, and $q$ [m s$^{-1}$] is, like above, the surface runoff velocity.



## 2.5 River Sediment-Transport Model

We consider two types of sediment: suspended sediment in the aqueous phase (mobile component) and bed sediment (immobile component). Fig. 2 shows a schematic of the river sediment-transport model, which considers advection, dispersion, deposition, bank erosion, bed erosion, and lateral input of suspended sediments. We use this model to calculate

5 the average concentration of the mobile component and the mass of the immobile component for every computation cell (formed by two cross-sections) every hour.

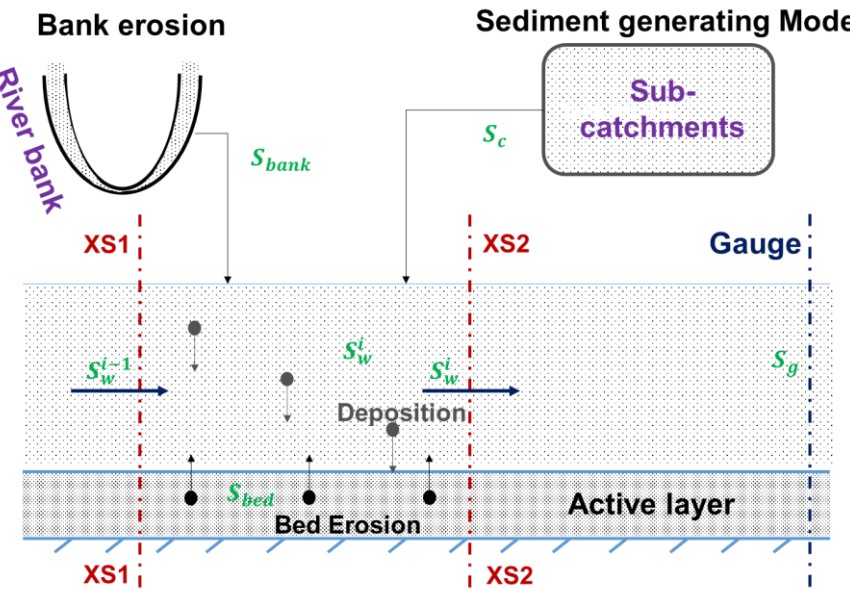

**Figure 2:** In-stream processes of the river suspended-sediment transport model considering deposition, bed erosion, bank erosion, and input from the catchment. XS1 and XS2 are the two cross sections bounding a cell in a Finite Volume scheme. $S_c$ and $S_{bank}$ are sediments
10 from the catchment and bank erosion. $S_{bed}$ indicates the bed sediment mass. $S_w^i$ stands for the concentration of suspended sediments in the i-th cell. $S_g$ is the suspended-sediment concentration at a river gauge.

## (1) Mobile Component

We use a Finite Volume discretization for suspended-sediment transport, considering storage in the aqueous phase, advection, dispersion, bed and bank erosion, deposition, and lateral inputs:

$$\frac{\partial(c_w V)}{\partial t} = -\frac{\partial(c_w Q)}{\partial x}\Delta x + AD\frac{\partial^2 c_w}{\partial x^2}\Delta x + (r_{bed} + r_{bank})\Delta x - r_d V + \sum c_{lat}^i Q_{lat}^i \tag{7}$$

in which $c_w$ [mg L$^{-1}$] is suspended-sediment concentration; $V$ [m$^3$] is the cell volume; $\Delta x$ [m] is the cell length; $Q$ [m$^3$ s$^{-1}$] and $A$ [m$^2$] are the flow rate and cross sectional area; $D$ [m$^2$ s$^{-1}$] is the dispersion coefficient; $c_{lat}^i$ [mg L$^{-1}$] and $Q_{lat}^i$ [m$^3$ s$^{-1}$] represent the suspended-sediment concentration and flow rate of the $i$-th lateral inflow; $r_d$ [mg L$^{-1}$ s$^{-1}$], $r_{bank}$ [g m$^{-1}$ s$^{-1}$], and $r_{bed}$ [g m$^{-1}$ s$^{-1}$] indicate the deposition, bed-erosion, and bank-erosion rates, respectively. For the advective term, we use





upstream weighting, whereas the second derivative of concentration appearing in the dispersion term is evaluated by standard Finite Differences.

**(2) Immobile Component**

For simplification, we account for one active layer only in the bed sediment per cell, and consider only the average grain size. Deposition of suspended sediments leads to a mass flux from the aqueous phase to the bed layer, whereas bed erosion causes a mass flux in the opposite direction:

$$\frac{\partial M_{bed}}{\partial t} = r_d \frac{V}{\Delta x} - r_{bed} \tag{8}$$

in which $M_{bed}$ [g m$^{-1}$] is the sediment mass per unit channel length in the active layer on the river bed.

**a. Deposition**

The deposition rate $r_d$ of particles can be calculated by (Krone, 1962):

$$r_d = \begin{cases} \left(1 - \frac{\tau_b}{\tau_e}\right) \frac{v_s c_w}{y} & \text{if } \tau_b < \tau_e \\ 0 & \text{otherwise} \end{cases} \tag{9}$$

in which $\tau_b$ [N m$^{-2}$] and $\tau_e$ [N m$^{-2}$] represent the bottom shear stress of the river and the threshold shear stress of particle erosion (see below); $y$ [m] denotes the water depth; and $v_s$ [m s$^{-1}$] is the settling velocity.

**b. Bed Erosion**

We consider two types of bed erosion, namely particle erosion and mass erosion, which correspond to two thresholds of the bottom shear stress. The bed erosion rate $r_{bed}$ can be calculated by (Partheniades, 1965):

$$r_{bed} = \begin{cases} M_{me} \left[\frac{\tau_b}{\tau_m} - 1 + \frac{M_{pe}}{M_{me}}\left(\frac{\tau_m}{\tau_e} - 1\right)\right] & \text{if } \tau_b > \tau_m \\ M_{pe}\left(\frac{\tau_b}{\tau_e} - 1\right) & \text{if } \tau_e < \tau_b \leq \tau_m \\ 0 & \text{otherwise} \end{cases} \tag{10}$$

in which $r_{bed}$ [g m$^{-1}$ s$^{-1}$] is bed erosion rate; $\tau_m$ [N m$^{-2}$] represents the mass erosion threshold; whereas $M_{pe}$ [g m$^{-1}$ s$^{-1}$] and $M_{me}$ [g m$^{-1}$ s$^{-1}$] denote the specific rates of particle and mass erosion.

**c. Bank Erosion**

In our model, the bank erosion rate $r_{bank}$ is calculated by:

$$r_{bank} = \begin{cases} \kappa \rho L y (\tau_{bank} - \tau_{bc}) & \text{if } \tau_{bank} > \tau_{bc} \\ 0 & \text{otherwise} \end{cases} \tag{11}$$

in which $\tau_{bank}$ [N m$^{-2}$] and $\tau_{bc}$ [N m$^{-2}$] are the bank shear stress and critical shear stress for bank erosion. $\kappa$ [m$^3$ N$^{-1}$ s$^{-1}$] is the erodibility coefficient. $\rho$ [kg m$^{-3}$] is density of bank material. $L$ [km] is length of the river bank.




## 3 Application to the Ammer Catchment, Germany

### 3.1 Study Area

We applied the integrated sediment transport model to the Ammer catchment, located in southwest Germany (Fig. 3). River Ammer is a tributary to River Neckar within the Rhine basin. It covers approximately 130 $km^2$, dominated by agricultural

land use that accounts for 67.35 % of the total area. The hydrogeology is dominated by the middle-Triassic Upper Muschelkalk limestone formation which forms the main karstified aquifer (Selle et al., 2013). In this catchment, annual precipitation is 700-800 mm. The Ammer River, approximately 12 km long, is the main stem with a mean discharge of ~1 $m^3$ $s^{-1}$. It has two major tributaries, the Kochhart and Käsbach streams. Two wastewater treatment plants (WWTPs), Gäu-Ammer and Hailfingen, also contribute flow and suspended sediments to the Ammer River. During dry weather conditions,

the discharge of WWTP Gäu-Ammer is 0.10–0.12 $m^3$ $s^{-1}$, and the effluent turbidity is approximately 3 NTU (Nephelometric Turbidity Units). The WWTP in Hailfingen is comparatively small with flow rates of 0.012–0.015 $m^3$ $s^{-1}$, and its turbidity is in the same range as that of the WWTP Gäu-Ammer. Only one gauging station is installed in the main channel of the Ammer River at the outlet of the studied catchment (red triangle in Fig. 3); here, hourly discharge and turbidity measurements are available. Based on the digital elevation model (DEM) of the Ammer catchment, we delineated 14 sub-catchments using the

watershed delineation tool of the Better Assessment Science Integrating point & Non-point Sources (BASINS) model (see Fig. 3). Table 1 shows the proportions of different land-use types and the areas of each sub-catchment.

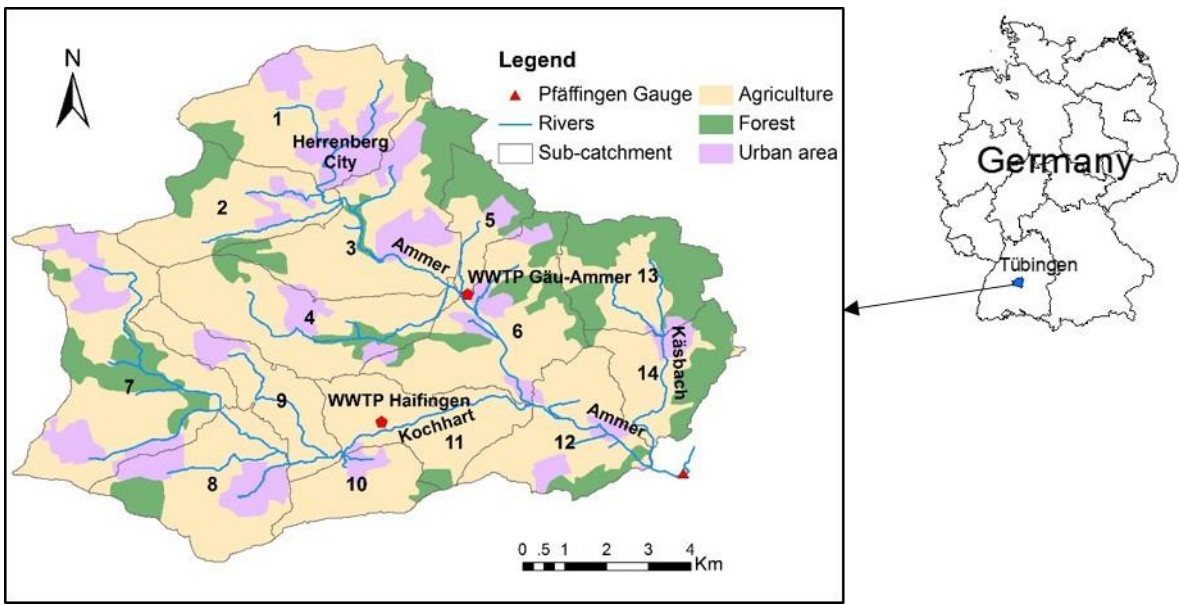

**Figure 3:** Location of the Ammer catchment and its sub-catchments, rivers and land-use. The numbers show identifiers (ID) of 14 sub-catchments that are characterized in more detail in Table 1. Two red regular pentagons represent two WWTPs in the study domain. The
20 red triangular indicates the gauge at the catchment outlet.





**Table 1.** Properties of the Ammer sub-catchments.

| ID of sub-catchment | Area of sub-catchment [km²] | Urban Area [km²] | Agriculture* [km²] | Forest [km²] |
|---|---|---|---|---|
| 1 | 12.70 | 3.78 | 7.80 | 1.13 |
| 2 | 8.13 | 0.70 | 6.06 | 1.38 |
| 3 | 13.53 | 2.47 | 8.13 | 2.92 |
| 4 | 11.15 | 1.19 | 8.70 | 1.25 |
| 5 | 3.97 | 0.46 | 1.62 | 1.89 |
| 6 | 11.80 | 1.53 | 7.69 | 2.59 |
| 7 | 17.12 | 3.30 | 10.65 | 3.16 |
| 8 | 10.10 | 2.41 | 6.74 | 0.95 |
| 9 | 6.14 | 0.66 | 5.48 | 0.00 |
| 10 | 4.55 | 0.50 | 3.87 | 0.18 |
| 11 | 7.74 | 0.05 | 7.39 | 0.30 |
| 12 | 8.66 | 1.04 | 6.73 | 0.89 |
| 13 | 8.36 | 0.21 | 3.39 | 4.76 |
| 14 | 6.60 | 0.58 | 3.66 | 2.35 |
| Area of land use [km²] | 130.54 | 18.87 | 87.92 | 23.75 |
| Proportion of land use [%] | 100 | 14.45 | 67.35 | 18.19 |

*The agricultural land in the Ammer catchment is dominated by non-irrigated arable land (80.2 % of the total agricultural areas), the crop of which is mainly cereals with annual rotation, and complex cultivation land (e.g., vegetables, 17.5 %). The rest (2.3 %) is principally agricultural area with natural vegetation. Therefore, we summarize the three types of arable land and use the same parameterization to estimate soil erosion.

### 3.2 Data Sources

Hourly precipitation and air-temperature data are the driving forces of the hydrological model. We use hourly precipitation data of the weather station Herrenberg, operated by the German weather service DWD (CDC, 2017), whereas air temperatures are taken from the weather station Bondorf of the agrometerological service Baden-Württemberg (BwAm, 2016). The generation and transport of sediments behave differently for different land use and topography. We use the digital elevation model with 10m resolution and land-use map of the state topographic service of Baden-Württemberg and Federal Agency for Cartography and Geodesy (BKG, 2009;LUBW, 2011). The river-hydraulics model requires bathymetric profiles of River Ammer and its main tributaries. We use 230 profiles at 100 m spacing, obtained from the environmental protection agency of Baden-Württemberg (LUBW, 2010). Discharge and suspended-sediment concentration at the gauge Pfäffingen are used for model calibration and validation. The hydrograph was converted by rating curves, whereas the suspended sediment concentrations are derived from continuous turbidity measurements (Rügner et al., 2013). The water levels and turbidity data were measured by online probes (UIT GmbH, Dresden, Germany).





### 3.3 Parameter Estimation

For the estimation of parameters, we used the well-known Nash-Sutcliffe Efficiency (NSE) as model performance criterion:

$$\text{NSE} = 1 - \frac{\sum_{i=1}^{n}(O_i - M_i)^2}{\sum_{i=1}^{n}(O_i - \bar{O})^2} \tag{12}$$

in which $O_i$ and $M_i$ are the $i$-th observed and modelled values, $\bar{O}$ is the mean of all observed values. A large NSE-value,

indicates good agreement between model and data, whereas NSE-value smaller than zero imply that the model performs worse than taking the mean of all observations. We obtained the best set of parameters by systematically scanning the parameter space.

The hydrological model was applied to 14 sub-catchments. Each sub-catchment has three types of land use: agricultural areas, forest, and urban areas. We used daily average discharge data of 2013–2014 and 2015–2016 for calibration and validation, respectively. By Latin Hypercube Sampling (LHS) we generated 1000 realizations of the 14 parameters for calibration and calculated the corresponding NSE-value for each parameter set. If NSE was ≥ 0.55, the parameter set was regarded acceptable. In the same way, we used the accepted parameter sets for validation. Then, these sets were used to calculate 90 % confidence intervals and identify the best-fit parameter values.

For the calibration and validation of the sediment generating and the river sediment-transport models, we performed a literature survey to identify the expected range of each parameter. Then the parameter values were identified manually based upon the reference range and optimized by fitting the modelled and measured suspended sediment concentrations at the river gauge. All parameters are listed in Tables 2 and 3 for the sediment-generating model and the river sediment-transport model, respectively. Suspended-sediment concentrations were estimated from turbidity. The linear relationship between suspended-sediment concentrations and turbidity is robust in the Ammer River (Rügner et al., 2013;Rügner et al., 2014). Based on the updated measurements, the conversion factor of 2.02 (mg L[-1] NTU[-1]) was used to convert turbidity to suspended-sediment concentrations. We also observed turbidity values of ~3 NTU for the periods without runoff events. Many studies have shown that karst systems can contribute suspended sediments (Bouchaoua et al., 2002;Meus et al., 2013). The study of Rügner et al. (2013) also showed that karst springs in the Ammer catchment contribute to turbidity. For the Ammer River, the subsurface flow through the karst system dominates the river flow in periods without rainfall events. Thus, the turbidity under base-flow conditions is potentially generated by subsurface flow through the karst matrix.



**Table 2.** Parameters of the sediment-generating model

| Parameter symbol | Definition | Unit | Range | Reference | Value |
|---|---|---|---|---|---|
| $M_{max}$ | Maximum accumulation load | g m$^{-2}$ | 7.5–50 | (Piro and Carbone, 2014;Modugno et al., 2015;Bouteligier et al., 2002) | 23 |
| $k$ | Accumulation rate constant | d$^{-1}$ | 0.16–0.46* | (Rossman and Huber, 2016) | 0.33 |
| $K_w$ | Wash-off coefficient | d$^{0.5}$ m$^{-1.5}$ | 50–500** | (Rossman and Huber, 2016) | 80 |
| $n_w$ | Wash-off exponent | - | 0–3 | (Wicke et al., 2012;Modugno et al., 2015;Rossman and Huber, 2016) | 1.5 |
| $C_h$ | Proportionality constant | s m$^{-1}$ | 0.003–0.05 | (Gilley et al., 1993) | 0.001 |
| $\tau_c$ | Critical non-urban shear stress | N m$^{-2}$ | 0–21*** | (Bones, 2014) | 0.3 |

*The range of $k$ is calculated under the assumption that it takes 5–30 days to reach 90 percent of the maximum buildup;

**The range of $K_w$, 50–500 (1–10, U.S. units), is sufficient for most urban runoff;

***It is for the very erodible to resistant soil, not including very resistant soil.

**Table 3.** Parameters of the river sediment-transport model

| Parameter symbol | Definition | Unit | Range | Reference | Value |
|---|---|---|---|---|---|
| $v_s$ | Settling velocity | m s$^{-1}$ | 10$^{-6}$–10$^{-4}$* | (Brunner, 2016) | 4×10$^{-6}$ |
| $\tau_e$ | Particle erosion threshold | N m$^{-2}$ | 0.1–5 | (Winterwerp et al., 2012) | 2.5 |
| $\tau_m$ | Mass erosion threshold | N m$^{-2}$ | >$\tau_e$ | (Partheniades, 1965;Brunner, 2016) | 3.5 |
| $M_{pe}$ | Particle erosion rate | kg m$^{-1}$ d$^{-1}$ | 0.8–43.2 | (Winterwerp et al., 2012) | 30 |
| $M_{me}$ | Mass erosion rate | kg m$^{-1}$ d$^{-1}$ | >$M_{pe}$ | (Partheniades, 1965;Brunner, 2016) | 40 |
| $\kappa$ | Erodibility coefficient | m$^3$ N$^{-1}$ d$^{-1}$ | 0.0001–0.32 | (Clark and Wynn, 2007;Hanson and Simon, 2001) | 0.0018 |
| $\tau_{bc}$ | Critical bank shear stress | N m$^{-2}$ | 0–21.91 | (Clark and Wynn, 2007) | 5 |
| $\rho$ | Density of bank material | kg m$^{-3}$ | 2190-2700 | (Clark and Wynn, 2007) | 2650 |

*This range is calculated for the suspended sediment with average diameter 1–50 μm;

## 4 Results and Discussion

### 4.1 Quality of Model Calibration and Validation

10    The best-fit parameter set of the hydrological model resulted in NSE values of 0.63 and 0.59 for calibration and validation, respectively. Fig. 4 shows the measured and simulated hydrographs for the calibration and validation periods with 90 % confidence intervals. It can be seen that the discharge was reproduced quite well, both in the general trend and the dynamics. The measured discharge data almost all fall within the 90 % confidence interval of the simulation. Only a few events cannot be reproduced by the model. These events occurred in the summer months and probably resulted from thunderstorms, which



are very local and may be missing in the precipitation measurements, so that the resulting flow peaks could not be predicted by the hydrological model.

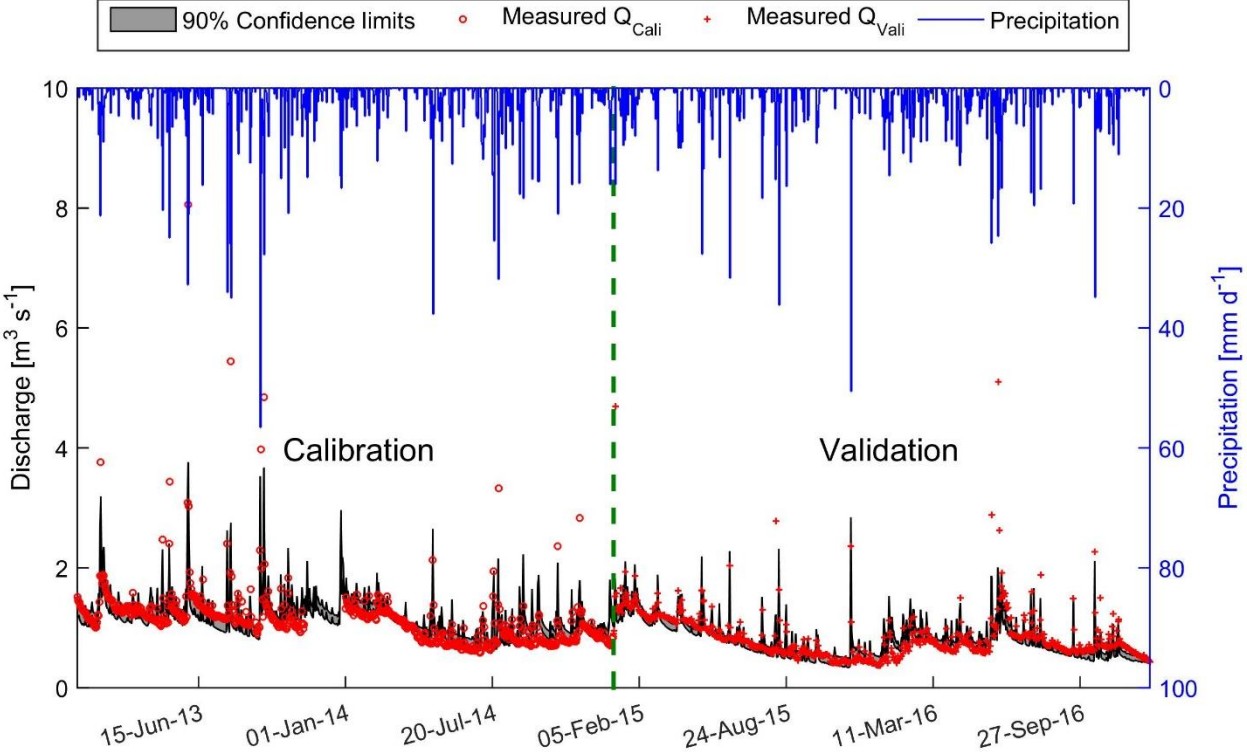

**Figure 4:** Calibration (left, year 2013–2014) and validation (right, year 2015–2016) of hydrological model, $Q_{Cali}$ and $Q_{Vali}$ are measured discharges used for calibration and validation, respectively.

Figure 5 depicts measured suspended-sediment concentrations and the simulation results of the sediment-transport model during the calibration (year 2014) and validation (year 2016) periods. The corresponding NSE values are 0.46 and 0.32, respectively, which indicates an acceptable fit, albeit not as good as for the hydrograph. The integrated sediment transport model can capture the dynamics of the suspended sediment concentrations. Especially, the model captures the concentration peaks well. However, two events with very high suspended sediment concentrations, one in the calibration and the other in the validation period, were not well fitted. These are events which were also not captured by the hydrological model, occurring in the summer months and due to thunderstorms.

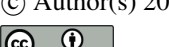



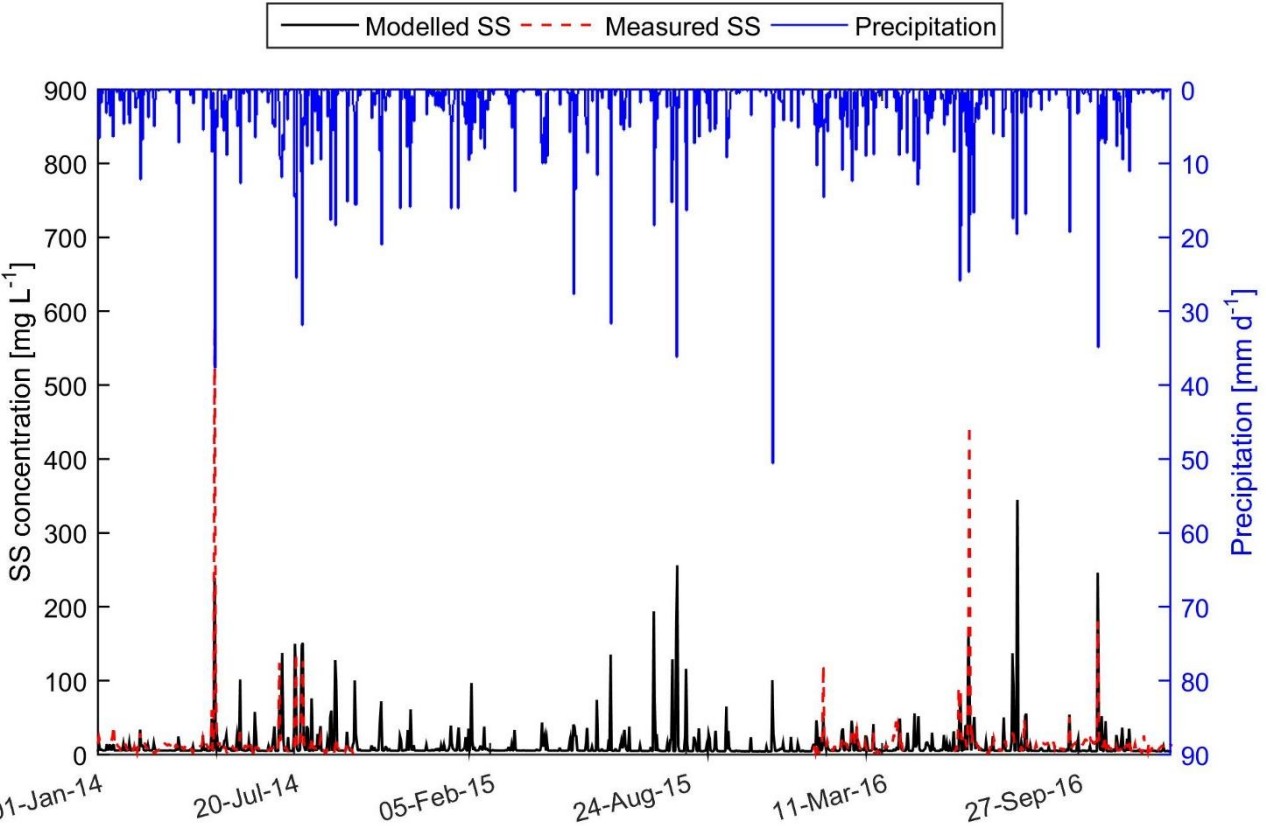

**Figure 5:** Modelled and measured suspended sediment concentrations used for calibration (year 2014) and validation (year 2016) of the sediment transport model. A data gap exists for year 2015.

### 4.2 Annual and Monthly Suspended Sediment Loads from Different Processes

5    Figure 6 displays the modelled annual suspended sediment loads from catchment and in-stream processes for the entire Ammer River network. The annual suspended sediment load at the gauge ranges between 410 and 550 ton $yr^{-1}$. Equation 13 describes the overall mass balance of sediments in the entire catchment:

$$Load_{gauge} =$$

$$(Load_{urb} + Load_{non-urb} + Load_{kar})_{Catchment} + (Load_{bde} + Load_{bke} - Load_{dep} - \Delta S)_{Stream} \qquad (13)$$

in which $Load_{gauge}$ [ton $yr^{-1}$] indicates the suspended-sediment load at the river gauge. $Load_{urb}$ [ton $yr^{-1}$], $Load_{non-urb}$

10   [ton $yr^{-1}$], and $Load_{kar}$ [ton $yr^{-1}$] denote the suspended-sediment loads from urban areas generated by surface runoff and WWTP effluent, non-urban areas generated by soil erosion, and karst system carried by subsurface flow, respectively. These





three terms represent catchment processes. $Load_{bde}$ [ton yr$^{-1}$], $Load_{bke}$ [ton yr$^{-1}$], $Load_{dep}$ [ton yr$^{-1}$], and $\Delta S$ [ton yr$^{-1}$] are the suspended-sediment loads from bed erosion, bank erosion, deposition, and the change of sediment storage in the entire river channel, respectively. These four terms represent the in-stream processes.

In the Ammer catchment, urban particles (266–337 ton yr$^{-1}$) and the sediment input from the karst system (106–160 ton yr$^{-1}$) dominate the annual suspended sediment load, accounting for 59.1 % and 24.9 %, respectively. Bed erosion, bank erosion, and non-urban sediment contribute much less, namely 6.2 %, 6.3 %, and 3.5 % of the total annual load, respectively. The contribution of non-urban runoff sediment in the Ammer catchment was very small. During the simulation period, annual precipitation was low, averaging 700 mm per year and there were only few events with the precipitation intensity excessing 10 mm h$^{-1}$, while infiltration rates of loamy soil and clay loamy soil are 10–20 mm h$^{-1}$ and 5–10 mm h$^{-1}$, respectively. Thus, hardly any surface runoff can occur in the non-urban area, so that sediment generation and transport from non-urban areas to the river channel are small.

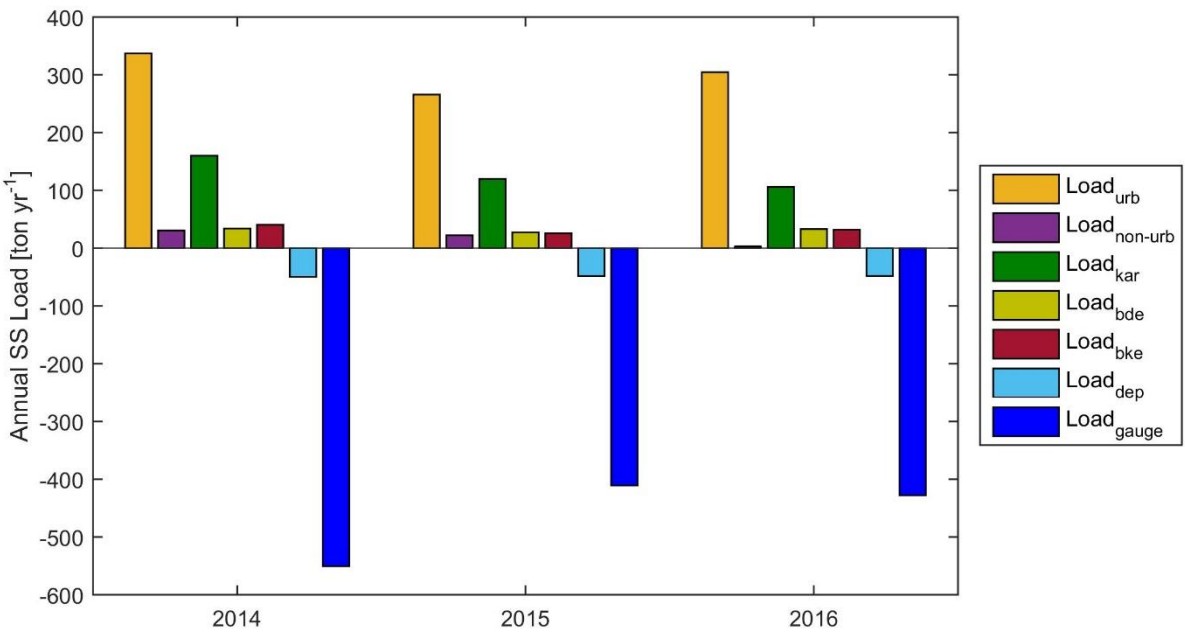

**Figure 6:** Annual suspended sediment loads from different processes. $\boldsymbol{Load_{gauge}}$ is calculated by modelled discharge and suspended sediment concentrations at catchment outlet. $\boldsymbol{Load_{urb}}$, $\boldsymbol{Load_{non-urb}}$, and $\boldsymbol{Load_{kar}}$ are calculated using the results of sediment generating model. $\boldsymbol{Load_{dep}}$, $\boldsymbol{Load_{bde}}$, and $\boldsymbol{Load_{bke}}$ are the sum of deposition load, suspended sediments eroded from river bed and river bank of the entire river network for a whole year, respectively. In this figure, the positive values represent sediment input to the river channel, while negative values denote sediment output from the river channel.

To identify seasonal variations of suspended sediment loads originating from different processes, we used the model results of 2014–2016 to analyze the monthly mean suspended sediment loads from the urban areas, non-urban areas, karst system, bed erosion, bank erosion, and deposition (Fig. 7). More suspended-sediment loads from urban areas and at the




gauge can be observed in June and July (summer months). In summer months the rain intensity is normally higher, which results in higher discharge, more sediments generated in urban areas, and higher suspended-sediment loads at the gauge. Monthly suspended-sediment loads at the gauge have similar dynamics as the monthly urban particle contributions. The suspended-sediment load from the karst system is higher in winter months because the subsurface flow in the Ammer

catchment is higher in winter months. Non-urban particles contribute during few months only to the overall particle flux since annual precipitation and rainfall intensity were relatively small so that surface runoff generated from non-urban areas was also low.

In the model simulation period, the seasonal patterns of bed erosion and bank erosion are obvious. High bed erosion and bank erosion occur from June to August due to increased bed shear occurring during big events. The area above the line of

$Load_{gauge}$ indicates the deposition, which shows small variations with a slight increase in July and August. The slight increase in summer is due to increased suspended-sediment concentrations during summer months. Comparing monthly mean bed erosion and deposition shows that bed erosion was greater than deposition in July, which indicates that accumulated bed sediment can be partly eroded in July.

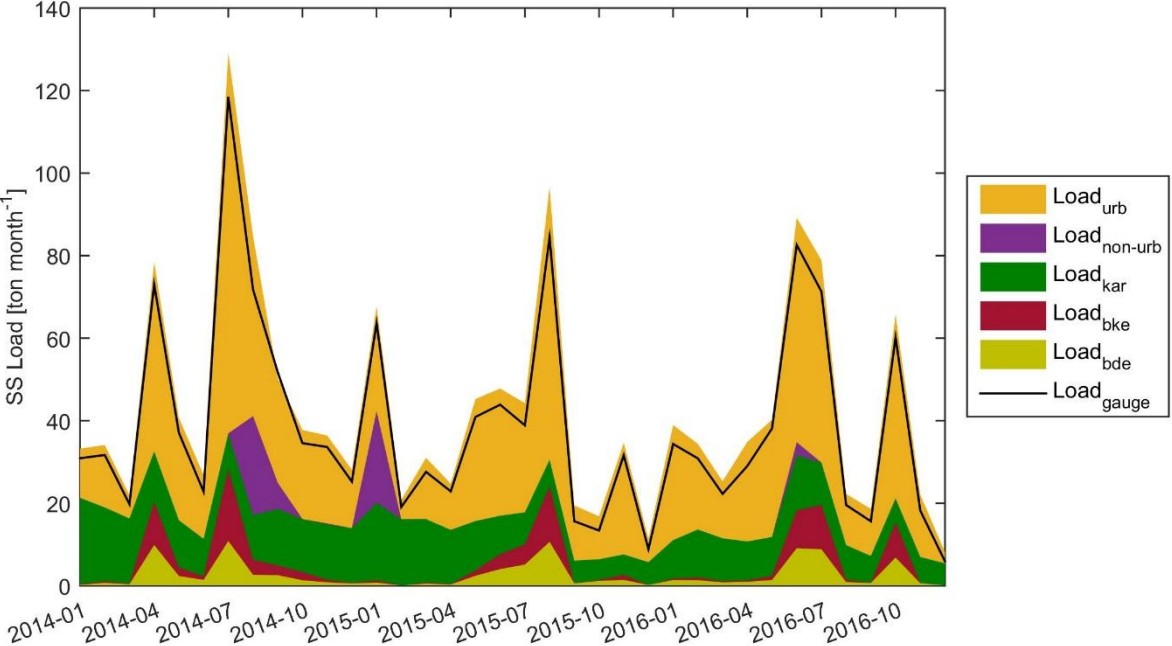

**Figure 7:** Monthly mean suspended-sediment load from different processes, calculated using the model results of 2014-2016. $Load_{gauge}$, $Load_{urb}$, $Load_{non-urb}$, and $Load_{kar}$ are the monthly mean suspended-sediment load at the gauge and from urban areas, non-urban areas, and karst system. $Load_{bde}$ and $Load_{bke}$ represent monthly mean suspended-sediment load from bed erosion and bank erosion for the entire river network, respectively. The area above the line of $Load_{gauge}$ is the monthly mean deposition, $Load_{dep}$.



### 4.3 Suspended-Sediment Sources under Different Flow Conditions

The relationship between hourly mean discharge and hourly suspended sediment loads from catchment, bed erosion, and bank erosion is demonstrated by the model simulations (Fig. 8). The hourly suspended-sediment load from catchment monotonically increases with increasing hourly mean discharge by a power-law relationship (Fig. 8a), which is consistent with the particle wash-off rate being a power function of discharge. Bed erosion requires that the bed shear stress exceeds a critical value, so that bed erosion is almost 0 when hourly mean flow is smaller than 1.5 $m^3$ $s^{-1}$ (Fig. 8b). For discharge larger than this threshold (1.5 $m^3$ $s^{-1}$), bed erosion increases approximately linearly with discharge. The simulated hourly bed-erosion loads for a given flow rate vary substantially because bed erosion is not only influenced by the shear stress, which directly depends on discharge, but also on the bed sediment storage, which depends on previous deposition and erosion events. Bank erosion occurs when the hourly mean flow rate is larger than 2.5 $m^3$ $s^{-1}$ (Fig. 8c). The relationship between bank-erosion related loads and discharge is more unique than that of bed-erosion loads because we assume an infinite source for bank erosion.




**Figure 8:** Relationship between simulated hourly mean flow and hourly suspended-sediment loads from the catchment (a), bed erosion (b), and bank erosion (c), in which bed erosion and bank erosion are sums over all computation cells. Loads from catchment is the sum of contributions from urban areas, non-urban areas, and karst system.

5     Figure 9 shows the suspended sediment loads from in-stream (bed erosion and bank erosion) and catchment processes (input from karst system, urban areas, and non-urban areas) under different flow regimes. The fractions of suspended-



sediment contributions from different processes change with flow regimes. The contributions of in-stream processes are negligible in the flow regime of discharge smaller than 5 m³ s⁻¹. With the discharge increasing, the contributions of in-stream processes increase. The in-stream processes play significant roles in high flow regimes, which contribute 23 % and 34 % of total suspended sediment loads under flow regimes of $10 \leq Q$ [m³ s⁻¹] < 15 and Q [m³ s⁻¹] ≥15, respectively. The relative

5    contribution of karst system is high in the low flow regime (Q [m³ s⁻¹] < 5), while it can be neglected under high flow regimes (Q [m³ s⁻¹] > 10). With the increase in flow rates, the contribution of urban particles become dominant in terms of catchment processes, especially when discharge is larger than 10 m³ s⁻¹.

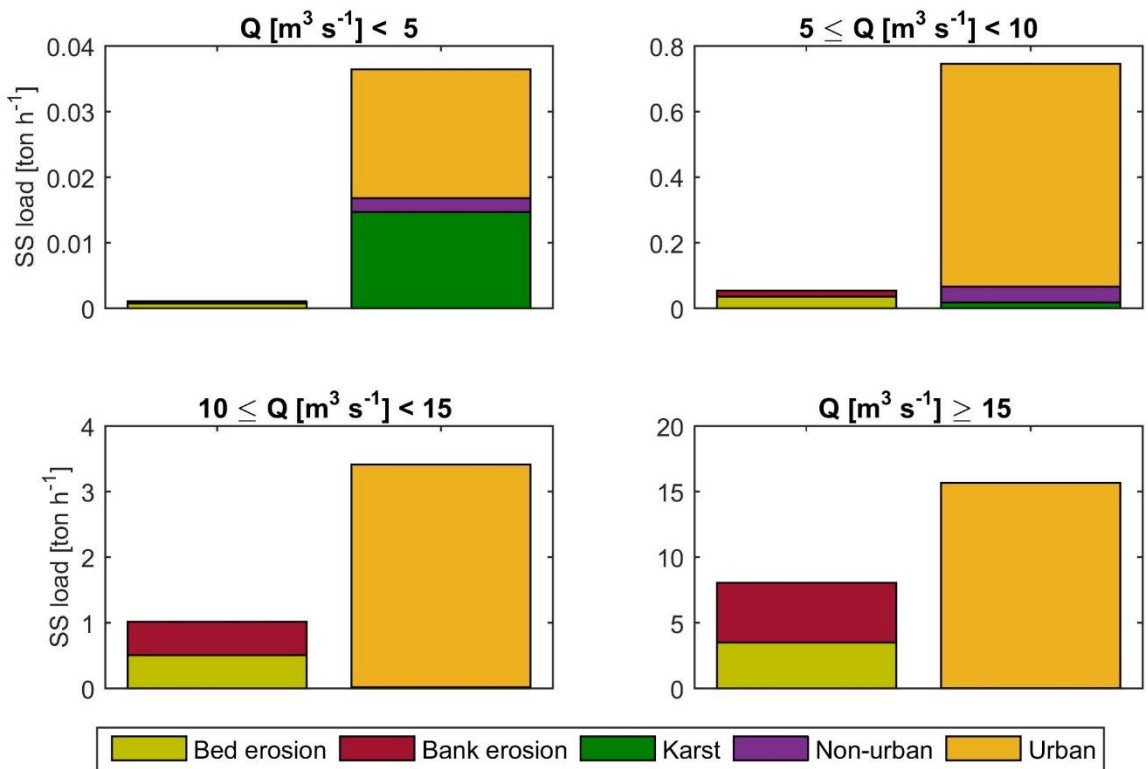

**Figure 9:** Simulated suspended sediment load from bed erosion, bank erosion, karst system, non-urban areas, and urban areas (including
10    suspended sediment from WWTPs) under different flow regimes, the suspended sediment loads are the mean values for the specific flow regimes.

From above observations, we can see that the sources of suspended sediments differ under different flow conditions in the following way:

| | |
|---|---|
| Q [m³ s⁻¹] < 1.5: | Suspended sediment load is dominated by contributions from the catchment (karst system, non-urban areas, and urban areas), while bed erosion and bank erosion can be neglected; |
| $1.5 \leq Q$ [m³ s⁻¹] < 2.5: | Bed erosion starts contributing; |



| | |
|---|---|
| $2.5 \leq Q \ [m^3 \ s^{-1}] < 5$: | Bank erosion starts contributing, but the contributions from bed and bank erosion are still negligible. Contributions from urban areas and karst system are dominant; |
| $5 \leq Q \ [m^3 \ s^{-1}] < 10$: | Bed and bank erosion contributes more, but the major contribution is still from catchment, especially from urban areas. Bed erosion contributes less than 5 % and bank erosion contributes less than 3 %. The relative contribution from karst system becomes very small; |
| $Q \ [m^3 \ s^{-1}] \geq 10$: | Suspended sediment contributions from bed and bank erosion are significant. The contribution of in-stream processes can be up to 35 % of the total suspended sediment load when discharge is larger than 15 m³ s⁻¹. The contribution from urban areas is largest, which dominates the catchment input. |

**4.4 Hotspots and Hot Moments of Bed Erosion in the Ammer River**

The annual mean rates of bed erosion and deposition (mass per unit length per year) along the main channel can be used to identify hotspots of bed erosion and net sediment trapping (Fig. 10). The rates of deposition and bed erosion vary substantially along the main stem, ranging from essentially zero to a maximum of 8.6 kg m yr⁻¹ and 8.0 kg m yr⁻¹,
respectively. Bed erosion is higher in the river segment close to the gauge because the flow rate is higher due to the contributions of tributaries. Bed erosion is rather low in the river segments of 5–6.5 km, 7–8 km, 8.5–9 km, and 10–11 km to the gauge, where the channel slope is very mild. The river sections with the steepest channel slope typically don't show the highest bed erosion because there is not enough sediment available for erosion, which is caused by insufficient deposition. Fig. 10 also shows that when the channel slope is very mild, the deposition rate is very high, while the bed erosion rate is
nearly zero. These are sections where net sediment trapping (grey shaded areas) was observed. With the channel slope increasing, bed erosion rates increase and deposition rates decrease. In a small range of channel slopes, deposition rates are equal to erosion rates, resulting in a local equilibrium. If the channel slope continues increasing, the erosion rate will be higher than the deposition rate, which results in net sediment erosion if there are available sediment storages in the channel (pink shaded areas, very few in Fig. 10). Where the channel slope is very steep, both sediment deposition and erosion rates
are very small.



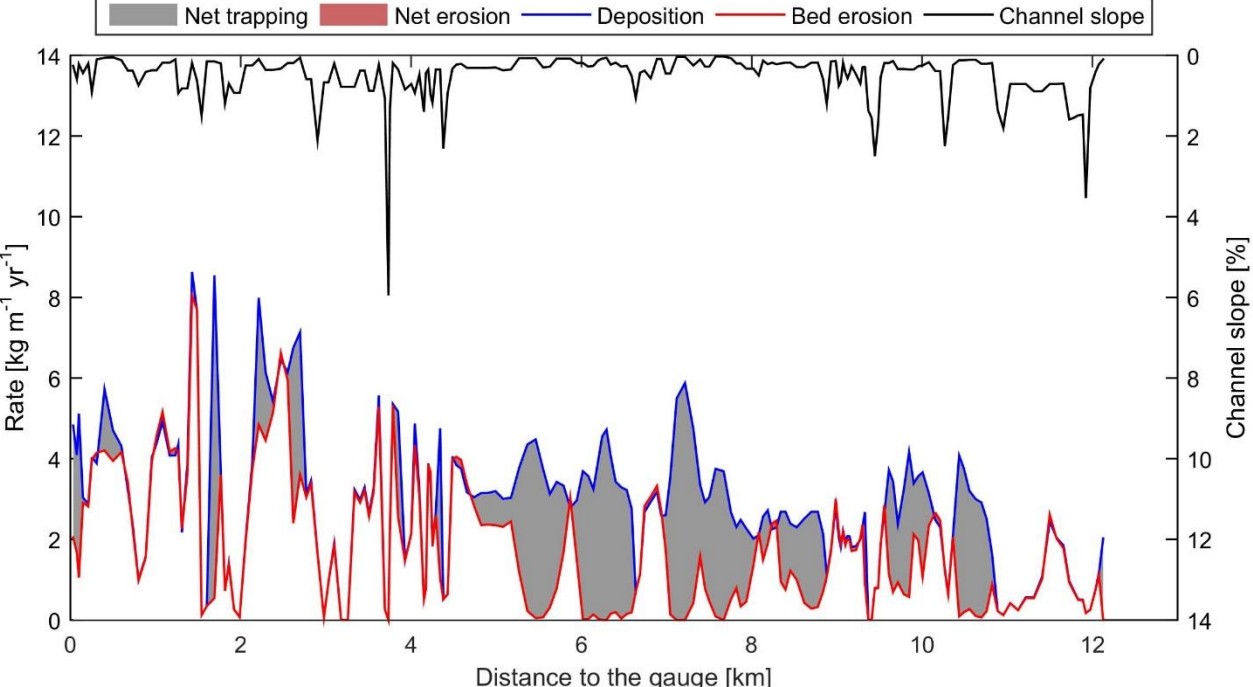

**Figure 10:** The distribution of the annual mean deposition, bed erosion, net sediment trapping, net sediment erosion, and channel slope along the main channel of the Ammer River (flow direction from right to left). The grey and pink shaded areas represent net sediment trapping and erosion, respectively.

5     Figure 11 shows monthly means of the bed erosion rates along the Ammer main stem, computed for the simulated years 2014 to 2016. Bed erosion is stronger in the summer months, especially in July, which is consistent with the monthly load of suspended sediments discussed in Sect. 4.2. The hot moments of bed erosion are the extreme events caused by summer thunderstorms. The downstream river segments close to the gauge show higher bed erosion rates than the sections further upstream because flow rates and thus bed shear stresses are higher even with identical channel slope.





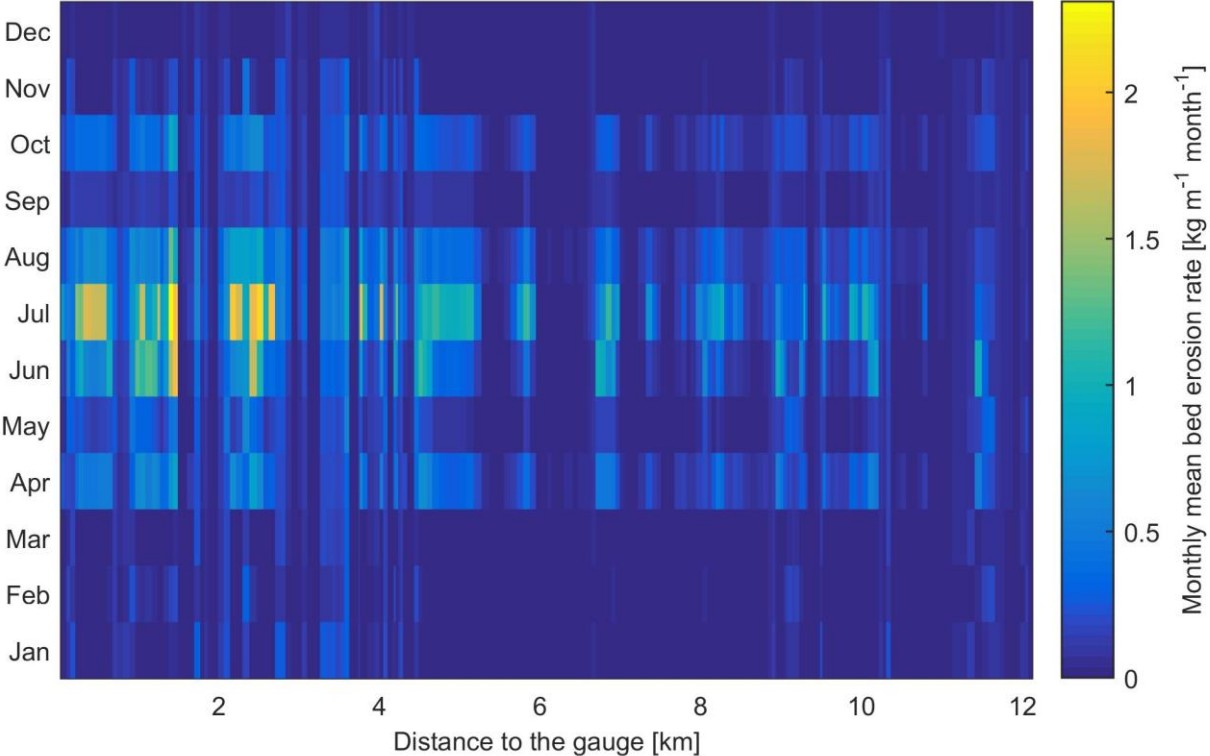

**Figure 11:** Monthly mean bed erosion along the channel of the Ammer River upstream of the gauge (flow direction from right to left).

**5 Conclusions**

Suspended sediment transport is of great importance for river morphology, water quality, and aquatic ecology. In this study, we have presented an integrated sediment-transport model, combining a conceptual hydrological model with a river-hydraulics model, a model of sediment generation, and a shear-stress dependent sediment-transport model within the river, which enables us to investigate the major contributors to the suspended-sediment loads in different river sections under different flow conditions.

In the dominantly groundwater-fed Ammer catchment, annual suspended-sediment load is dominated by the contributions of urban particles and sediment input from the karst system. The contribution from non-urban areas is small because the low annual precipitation and intensity in this region result in a very weak surface runoff from non-urban areas, thus very few non-urban particles are generated and transported to the river channel. In-stream processes, i.e. bed erosion and bank erosion play significant roles in high flow regimes ($Q > 10 \ m^3 \ s^{-1}$). The flow rate governs the contributions of different processes to the suspended sediment loads. Especially, bed erosion and bank erosion take place when flow rates reach the corresponding thresholds, $1.5 \ m^3 \ s^{-1}$ and $2.5 \ m^3 \ s^{-1}$, respectively. The channel slope has significant effects on the



deposition and bed erosion rates. Net sediment trapping was found in the river segments with very mild channel slopes in the Ammer River during the simulation period with events of a 2-year return period. Finally, the river hydraulics model is necessary to differentiate sediment sources and sinks of in-stream processes i.e. shear stress related deposition, bed erosion and bank erosion.

5    The model and results of this study are useful and essential for further research on the fate and sediment-facilitated transport of hydrophobic pollutants like PAHs, and for the design of optimal sampling regimes to capture the different processes that drive particle dynamics. In addition, the analysis of deposition and bed erosion in the Ammer stem provides information on the distribution of net sediment trapping within the channel, which would be a good indicator for channel dredging to improve water quality.

**Code availability**

The code is available on request.

**Appendix A. Catchment-Scale Hydrological Model**

The hydrological model in the integrated sediment transport model is composed of three storage zones in vertical direction with a quick recharge component and an urban surface runoff component. Detailed processes are shown below.

15    We applied this model to 14 sub-catchments of the study domain. Each sub-catchment includes three different land use: urban area, agriculture and forest. For urban area, we consider effective urban area such as roads and roofs and ineffective urban area such as parks and gardens. We use the same parameters of agriculture for ineffective urban area.

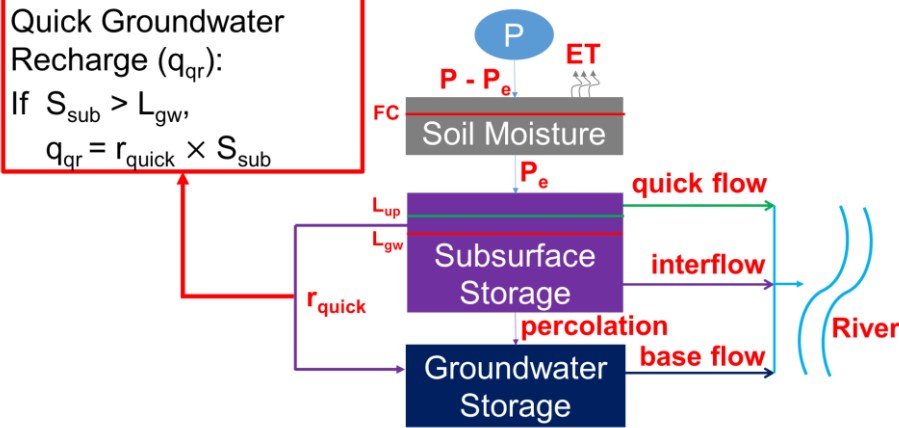

**Figure A1:** The hydrological model for the Ammer catchment with three storage zones (soil moisture, subsurface storage and groundwater storage), a quick groundwater recharge and an urban surface runoff component

The effective urban area is used for surface runoff component, the ratio is calculated by:


$$r_{eff} = \frac{A_{eff}}{A_{urb}} \qquad (A1)$$

in which $r_{eff}$ [-] is the ratio of effective urban area over total urban area, $A_{eff}$ [km$^2$] and $A_{urb}$ [km$^2$] represent areas of effective urban area and total urban area, respectively.

The effective precipitation to the subsurface storage for agriculture, forest and ineffective urban area is calculated below:

$$P_e = \left(\frac{SM}{FC}\right)^\alpha P \qquad (A2)$$

in which $P_e$ [mm d$^{-1}$] indicates effective precipitation, $P$ [mm d$^{-1}$] is precipitation, $SM$ [mm] and $FC$ [mm] are soil moisture and maximum soil storage capacity, respectively, $\alpha$ [-] is a shape factor.

We use long-term monthly mean evapotranspiration to calculate the actual evapotranspiration with a temperature adjustment.

$$L_{et} = FC c_{et} \qquad (A3)$$

$$ET_t = [1 + c_t(T - T_m)]ET_m \qquad (A4)$$

$$ET_a = \begin{cases} ET_t, & SM \geq L_{et} \\ \frac{SM}{L_{et}} ET_t, & SM < L_{et} \end{cases} \qquad (A5)$$

in which $L_{et}$ [mm] is a threshold for maximum evapotranspiration, $c_{et}$ [-] is a factor to calculate $L_{et}$. $ET_t$ [mm d$^{-1}$] represents the maximum evapotranspiration at temperature $T$ [℃]. $ET_m$ [mm d$^{-1}$] and $T_m$ [℃] indicate long-term monthly mean evapotranspiration and long-term monthly mean temperature, respectively, $c_t$ [℃$^{-1}$] is a temperature adjustment factor. $ET_a$ [mm d$^{-1}$] represents actual evapotranspiration, which reaches maximum evapotranspiration when soil moisture is greater than the threshold for maximum evapotranspiration. Otherwise, it increases linearly with soil moisture.

The top storage layer, soil moisture, is calculated by:

$$\frac{dSM}{dt} = P - P_e - ET_a \qquad (A6)$$

in which $\frac{dSM}{dt}$ [mm d$^{-1}$] represents the change rate of soil moisture. It is used for agriculture and forest. The change rate of soil moisture for urban area is $\frac{dSM}{dt}(1 - r_{eff})$, because we assume that precipitation on the effective urban area will directly become urban surface runoff.

The surface runoff in the effective urban area, overflow and interflow are calculated by:

$$q_{effurb} = P \qquad (A7)$$

$$q_{of} = \begin{cases} 0, & S_{up} < L_{of} \\ k_{of}(S_{up} - L_{of}), & S_{up} \geq L_{of} \end{cases} \qquad (A8)$$

$$q_{if} = k_{if} S_{up} \qquad (A9)$$





$$q_{bf} = k_{bf}S_{gw} \tag{A10}$$

in which $q_{effurb}$ [mm d$^{-1}$] is surface runoff in the effective urban area. $q_{of}$ [mm d$^{-1}$] represents overflow when subsurface storage $S_{up}$ [mm] is greater than an overflow threshold $L_{of}$ [mm]. It is used for agriculture, forest and ineffective urban area. $q_{if}$ [mm d$^{-1}$] represents interflow. $k_{if}$ [d$^{-1}$] is a rate constant. $q_{bf}$ [mm d$^{-1}$] represents base flow, $S_{gw}$ [mm] is groundwater storage, $k_{bf}$ [d$^{-1}$] is a base flow recession coefficient.

The two equations below are used to calculate percolation and quick recharge.

$$q_{perc} = k_{perc}S_{up} \tag{A11}$$

$$q_{qr} = \begin{cases} 0, & S_{up} < L_{qr} \\ k_{qr}(S_{up} - L_{qr}), & S_{up} \geq L_{qr} \end{cases} \tag{A12}$$

in which $q_{perc}$ [mm d$^{-1}$] represents percolation from soil moisture to subsurface storage. $k_{perc}$ [d$^{-1}$] is a rate constant. $q_{qr}$ [mm d$^{-1}$] represents quick recharge, which occurs when subsurface storage reaches a quick recharge threshold $L_{qr}$ [mm]. $k_{qr}$ [d$^{-1}$] is a rate constant.

The subsurface storage and groundwater storage are calculated by:

$$\frac{dS_{up}}{dt} = \begin{cases} P_e - q_{perc} - q_{qr} - q_{of} - q_{if}, & agriculture\ and\ forest \\ P_e(1 - r_{eff}) - q_{perc} - q_{qr} - q_{of} - q_{if}, & urban\ area \end{cases} \tag{A13}$$

$$\frac{dS_{gw}}{dt} = q_{perc} + q_{qr} - q_{bf} \tag{A14}$$

in which $\frac{dS_{up}}{dt}$ [mm d$^{-1}$] is the change rate of subsurface storage. In the urban area, only precipitation in the ineffective area can partly become recharge to the subsurface storage. $\frac{dS_{gw}}{dt}$ [mm d$^{-1}$] represents the change rate of groundwater storage.

**Competing interests**

The authors declare that they have no conflict of interest.

**Acknowledgements**

This study was supported by the German Research Foundation (Deutsche Forschungsgemeinschaft, DFG) within the Research Training Group "Integrated Hydrosystem Modeling" (grant GRK 1829). Additional funding is granted by the Collaborative Research Center SFB 1253 "CAMPOS – Catchments as Reactors".



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
