# Peer review of "Contributions of Catchment and In-Stream Processes to Suspended Sediment Transport in a Dominantly Groundwater-Fed Catchment"

_Hydrology and Earth System Sciences, 2018_

## Short Comment (SC1) · 21 Feb 2018

Though the demand on water use is ever increasing due to population growth, urbanization, and climate change, it is inevitable to accept the fact that the supply is limited. To compound this issue, the quality of water is also deteriorating due to human and natural activities. Among the activities that deteriorate the available water, sedimentation that is due to erosion, suspension, and deposition in rivers ranks at the top. Therefore, in this manuscript, the authors present an integrated sediment transport model to determine the sediment contribution from the Ammer River in Germany. Based on the review of this manuscript, the following points are highlighted:

[Figure]

1) The catchment input, bed erosion, and bank erosion increase with an increase in flow rates (See LN-18 P-1). Is this a generic statement? What is meant by catchment input? What is the expected relationship between the erosion and flow rate? What is mentioned in the literature? Is it possible to justify this statement (i.e., LN-18 P-1) using equation (10) and equation (11)?

2) Bed erosion and bank erosion are negligible when flow is smaller than the corresponding thresholds of 1.5 m3 s-1 and 2.5 m3 s-1, respectively (See LN-19 P-1). Is this statement about the rate? Moreover, the threshold values (i.e., 1.5 m3 s-1 and 2.5 m3 s-1) need to be normalized using some of the catchment properties to understand the authors' statement. The threshold value on bank erosion (i.e., 2.5 m3 s-1) is greater than the threshold value on bed erosion (i.e., 1.5 m3 s-1). What is mentioned in the literature?

3) Asper the authors, USLE and SEDD cannot estimate sediment generation by in-stream processes (See LN-15 P-2). Moreover, as per the authors, although SWAT/HSPF/HEC-RAS can simulate "physical processes", none of these models represent in-stream processes well, specifically in natural rivers (See LN-24 P-2). What are those instream processes? In think, the authors need to explain the way the sediment (e.g.., suspended) generation and transport is simulated in some of these models (e.g., SWAT) to understand the pitfalls of the existing models to solve the intended problem(s) in Germany.

4) The catchment-scale hydrological model is based on the HBV model (See LN-1 P-4). Does this statement need a reference? Moreover, the authors have added a quick recharge component and an urban surface runoff component to explain the special behavior of discharge in the Ammer catchment (See LN-4 P-4). The special behavior of discharge in the Ammer catchment and the reason(s) for including the additional components are not understood. Is the integrated sediment transport model applicable anywhere?

[Figure]

5) The HEC–RAS simulates hourly quasi-steady flow (See LN-15 P-4). What was the reason for not selecting the unsteady option in HEC-RAS? The details about the boundary conditions (e.g., Upstream/downstream) and initial states are missing in the manuscript. What types of boundary conditions you had in your model?

6) The distances between "computed" cross-sections range from 10 m to 100 m depending on the changes of river bathymetry (See LN-19 P-4). Are these the interpolated cross-section data. What was the interpolation algorithm? Did you use one of in-built(HEC-RAS) interpolation algorithms? Where did you have your observed cross-section data in your model? The details are missing in the manuscript. HEC–RAS model computes the hourly hydraulics for the all cross-sections of the main channel and two major tributaries of the Ammer River (See LN-18 P-4). Does this statement fit the section (i.e., model setup)?

7) The section 2.3 needs to be more detailed. Many details are missing in the manuscript. The modeled river schematization needs to be included in the manuscript. How did you include the tributaries in HEC-RAS? The details on the junctions/confluences are also needed.

8) The land use is classified into urban areas and non-urban areas (See LN-23 P-4). Impervious surfaces such as roads and roofs are regarded as urban areas, while non-urban areas consist of pervious surfaces such as gardens and "parks", agricultural areas, and forests (See LN-24 P-4). Does this statement contradict with your section 3.1(See Table 1)? Did you classify your LULC into urban and non-urban?

9) The sediment-generating model is used to obtain hourly sediments of different sources from the 14 sub-catchments (See LN-26 P-4). What is meant by different sources?

10) We use the urban-area algorithm of SWMM, which "performs well on particle buildup and wash-off for urban land use", to describe sediment generation from urban areas (See LN-28 P-4). Does this statement need a reference?

11) The variables in equation (1) need more description to understand the units (i.e., gm-2). The definitions of the variables need to include the area. Moreover, the variable M(t) is not found in the equation. Is equation (1) applicable only for urban areas? What is the reason(s)? Does the equation have a variable to show that it is applicable only for urban areas?

12) In equation (1), what is the value of "k" used in the model. What is the value of "Mmax" used in the model? The maximum buildup depends on the particle production and cleaning frequency, which is obtained through calibration (See LN-6 P-5). This statement needs more explanation.

13) In equation (5), what is the value of your critical stress? Does the value of critical stress vary with time? Don't you consider the particle accumulation in non-urban areas? Is equation (4) applicable for all non-urban areas? Moreover, sin(theta) is not the mean slope of the sub-catchment. Since theta is very small, you will end up saying sin(theta)=tan(theta)?

14) Is equation (11) formulated by the authors? Otherwise, the reference is required. In equation (11), what is the unit of bank erosion rate? This unit needs to be compared with the unit of bed erosion rate (i.e. equation (10)). Does the bank erosion rate vary spatially along the reach? Does equation (11) cover the bank erosion in the freeboard region? In equation (11), what is the value of your critical shear stress for bank erosion? Does the value of critical stress vary with time? In equation (11), what is the equation of your bank shear stress?

15) In equation (10), the units of bed erosion rate and the specific rates of particle and mass erosion are not understood. What is meant by "specific rate"? How do you compare this (i.e., specific rate) with the bed erosion rate? What are the values of the thresholds (i.e., mass erosion threshold and particle erosion threshold)? Does the bed erosion rate vary spatially along the reach?

16) Does equation (9) represent the "bank" and "bed" deposition rates? What is the

reason to condition based on the bottom shear stress of the river?

17) In equation (8), what is the assumption(s) made in formulating the first component of the right-hand-side of the equation?

18) Are your computations cells of equal size? As per LN-5 in P-6, the computation cells are formed by two cross-sections. However, your cross-sections are not equally spaced (See LN-19 P-4). Wont this influence your model outcome?

19) As per the title of the manuscript, the catchment is dominated by groundwater. However, the current version of the manuscript does not lead to understand this statement. Does the equations account in your suspended sediment transport model account for this statement (i.e., dominantly groundwater-fed catchment)?

20) The equation (7) needs to be derived from first principles. Does this equation account for sink (i.e., flow diversion)? Is this equation formulated correctly? Considering your equation (10) and equation (11), what is the unit of the third component in equation (7)? Did you use the equations (1-6) in your equation (7)? Which component of your equation (7) accounts for your equations (1-6)?

---

## Referee Comment (RC1) · Anonymous Referee #1 · 26 Feb 2018

The manuscript illustrates a coupled catchment-stream model for water quantity and sediment productions and transport developed for the Ammer River Basin (Germany), which has some important karstic contribution to baseline flow and suspended sediments. The physics- based model that is proposed includes a complex one-dimensional hydraulic component for calculating shear stress. Erosion rates are then based on shear stress concepts applied to erosion of bed and bank material (either deposited sediments or consolidate beds), as well as in-stream deposition. The model was developed to tackle Ammer Basin hydrology in particular, however it is proposed as an integrated model of general applicability. The model is built on components of other hydrological and sediment models. It appears to be very focused on in-stream processes. Conversely, sediment sources from land processes (soil erosion) seems too simplified. I have some concerns about the paper and its content. 1) first of all, it is not true that existing models do not account for in-stream processes (P2 L24). For example, I know that SWAT model offers several ways to tackle in-stream sediment transport and erosion, including some physics based approaches based on shear stress and the possibility to include cross section of reaches. Some literature has shown that these SWAT approaches work well. The authors should therefore revise their statements. 2) Sediments from urban land is modelled with a well-known wash off/build up approach. Instead erosion from non-urban areas is tackled with an (to my point of view overly) simplified approach (eqs. 4-6) whose main drivers are runoff and slope. Only one shear stress threshold is considered despite agricultural land diversity, which includes a variety of crops like cereals, vegetables and natural vegetation. This approach does not consider any variability in soil erodibility, or changes in crop cover during the year, which instead impact soil erosion from agricultural land especially among seasons. This flaw limits very much results drawn from the model especially in terms of seasonality and 'hot moments' of erosion. 3) I have some concerns about the calibration and validation of the model. The model has 14 calibration parameters and is calibrated vs 1 single station at the outlet of the Basin. I also note that calibrated parameter Ch (Table 2) which regulates the non-urban sediment loads, is lower than its initial range. The risk of over parameterization of this model is very high. Some sensitivity analysis should be shown and discussed as this represent a limit of potential conclusions of the paper. Calibration was driven by a LHS scheme but conducted manually. The authors state that calibration parameter sets were retained to derive 90% confidence intervals. However these confidence intervals are not shown nor further commented expect for a vague comment at P 11 L 14. The model runs at hourly time step. At what time step calibration and validation were conducted? Water discharge was calibrated for 2013-1014 and validated for 2015-2016. Sediments were calibrated for 2014 and validated for 2016. Why data for 2015 was not used in this exercise? Data was available as shown in fig 5 but model simulations are not shown. However, model simulations are

used for sediment balance considerations e.g. figures 6 and 7. Please explain. The model missed simulation of 2 large rainfall events (one in 2014 and one in 2016), where the highest sediment concentrations occurred. The explanation offered (P11 L 14, p12 L1-2) is that rainfall precipitation measurements 'may be missing'. This should be verified in the input data. In any case, these 2 events were the most important for sediment load, so all sediment balance is flaw as it cannot consider these main events. It would also be good to see some events in more details given the high temporal discretization of the model. 4) the results of the model indicate that urban land is the major source of sediments in the catchment. this is possible, but I find hard to believe that 67% of the basin (agricultural land) contributes almost nothing to sediment loads. Even if runoff production is very low and land is gentle sloping (P 21 Lines 10-12), I would expect more contribution. The authors should check with other lines of evidence (e.g. literature of soil erosion from agricultural land in the region) if their results are realistic. 6) Fig 8 indicates an increase of sediment sources following a power-law with discharge, which may make sense. However, I wonder if an excess of sediment transport capacity of the stream was considered in the model. This may regulate deposition when sediment sources are very high. I do not see this being considered in the model (but I may be wrong). Please discuss. 7) what data was used to set karstic sediment loads? 8) section 3.1 should precede model description. The model was built for the Ammer and some important information driving model conceptualization is given in this section, so this should come first. Information about measurement data should be given in this section. Please move P 9 Lines 14-17 and P10 Liens 18-25 to after current P 8 line 14. 9) please change color of Load urb and load bed in figs 6 and 7 to better distinguish them. 10) schematic text at P 18-19 lines 12 onward should be given as a table. 11) reference in the conclusion to events with 2-year return period (P 22 L 2) is surprising. No reference to return period is done before in the manuscript. Given that model failed to simulate two large rainfall events of the region, I find it hard to believe this statement.

42, 2018.

---

## Author Comment (AC1) · 8 Mar 2018

*We thank S. Mylevaganam for the detailed review and the constructive remarks.*
*In the following, the comments are set in normal fonts, and the responses are in italic.*

**Comments and Responses**

Though the demand on water use is ever increasing due to population growth, urbanization, and climate change, it is inevitable to accept the fact that the supply is limited.

[Figure]

To compound this issue, the quality of water is also deteriorating due to human and natural activities. Among the activities that deteriorate the available water, sedimentation that is due to erosion, suspension, and deposition in rivers ranks at the top. Therefore, in this manuscript, the authors present an integrated sediment transport model to determine the sediment contribution from the Ammer River in Germany. Based on the review of this manuscript, the following points are highlighted:

1. The catchment input, bed erosion, and bank erosion increase with an increase in flow rates (See LN-18 P-1). Is this a generic statement? What is meant by catchment input? What is the expected relationship between the erosion and flow rate? What is mentioned in the literature? Is it possible to justify this statement (i.e., LN-18 P-1) using equation (10) and equation (11)?

   *Response: By the catchment input we refer to the sum of sediments from urban areas, non-urban areas, karst system, and waste-water treatment plants (WWTP), that is, all sediments that are not generated by in-stream processes (bed and bank erosion).*

   *For a river with uniform cross section, we would expect a power-law relationship between the erosion and flow rate. Previous studies have shown that the bed load depends on stream power by a power-law function (Lammers and Bledsoe, 2018; Schneider et al., 2014), in which the stream power is a linear function of the flow rate for a given channel geometry. For the entire river system with non-uniform profiles along the reach, however, the cumulative erosion of the river could follow a different functional relationship on flow rate (in general, we expect that the erosion increases with the increase of flow rates, as the bottom shear stress monotonically depends on the flow rate). Equation (10) shows that bed erosion rate is a piecewise linear function of excess shear stress if the supply of bed sediments is infinite. Equation (11) indicates that the bank erosion rate increases linearly when the shear stress is greater than the threshold. Shear stress increases with the increase of flow rates. Therefore, the bed and bank erosion*

*increase with an increase in flow rates.*
*In the intended revision of the paper we will try to make these concepts clearer.*

2. Bed erosion and bank erosion are negligible when flow is smaller than the corresponding thresholds of 1.5 $m^3 s^{-1}$ and 2.5 $m^3 s^{-1}$, respectively (See LN-19 P-1). Is this statement about the rate? Moreover, the threshold values (i.e., 1.5 $m^3 s^{-1}$ and 2.5 $m^3 s^{-1}$) need to be normalized using some of the catchment properties to understand the authors' statement. The threshold value on bank erosion (i.e., 2.5 m3 s-1) is greater than the threshold value on bed erosion (i.e., 1.5 $m^3 s^{-1}$). What is mentioned in the literature?
*Response: Yes, the statement is about the rate. To normalize the threshold values of bed and bank erosion is a very good suggestion. We will use the mean discharge to normalize the thresholds and the changes will be shown in the revised manuscript. In our manuscript, we wanted to see the effects of flow rate on the total sediment erosion of the entire river. The result indicates a higher threshold of bank erosion than that of bed erosion, which is expected and reasonable. The bank material is more coherent than be bed sediments, thus requiring larger shear stress to induce bank erosion compared with bed erosion, which results in a higher threshold of flow rate for bank erosion. The literature shows a smaller critical shear stress for bed erosion (Winterwerp et al., 2012) than for bank erosion (Clark and Wynn, 2007).*

3. As per the authors, USLE and SEDD cannot estimate sediment generation by in-stream processes (See LN-15 P-2). Moreover, as per the authors, although SWAT/HSPF/HEC-RAS can simulate "physical processes", none of these models represent in-stream processes well, specifically in natural rivers (See LN-24 P-2). What are those instream processes? In think, the authors need to explain the way the sediment (e.g.., suspended) generation and transport is simulated in some of these models (e.g., SWAT) to understand the pitfalls of the existing models to solve the intended problem(s) in Germany.

*Response: The in-stream processes in the manuscript are deposition of sus-
pended sediment from the water phase to the river bed, bed erosion and bank
erosion due to excess shear stress. We have given some general information of
sediment generation of the listed models. We will provide more information on the
in-stream processes of the existing models in the revised manuscript, including
the limitations of SWAT.*

4. The catchment-scale hydrological model is based on the HBV model (See LN-1
P- 4). Does this statement need a reference? Moreover, the authors have added
a quick recharge component and an urban surface runoff component to explain
the special behavior of discharge in the Ammer catchment (See LN-4 P-4). The
special behavior of discharge in the Ammer catchment and the reason(s) for in-
cluding the additional components are not understood. Is the integrated sediment
transport model applicable anywhere?
*Response: We will add a reference to the HBV model. The reason for adding
a quick recharge component is that in the Ammer catchment, the measured hy-
drograph demonstrates a rapid increase in base flow in sporadic events. We
attribute this peculiar behavior to the hydrological functioning of karst with a deep
unsaturated zone (distance to groundwater up to 100m at the upstream end of
the catchment). In our conceptual model, we assume water storage in the deep
unsaturated zone, which spills over when a threshold value is reached, causing
quick groundwater recharge occur which then leads to a rapid increase of base
flow.
We have added an urban surface runoff component to obtain a surface runoff
depth in urban areas in order to simulate particle wash-off from urban land sur-
face. The integrated sediment model can be applied to other catchments with
characteristics similar to the Ammer catchment. In particular, the sediments are
mainly contributed by urban areas, surface runoff in the agricultural areas is so
weak and sporadic that the erosion in the agricultural area can be simplified, and*

*sediment production is driven by surface runoff rather than wind blow. Besides the karst-affected hydrology mentioned above, the Ammer catchment is special because the valley is too wide for the current stream flow. This stems from a different river system (River Nagold) cannibalizing the Ammer in the early Pleistocene. The valley has a size that fits to a stronger river that has lost a substantial fraction of its stream flow. This is why we observe so little erosion on the (too flat) hillslopes. While this situation is special, it is not unique. There are other rivers in South Germany that have lost their original stream flow in the course of the extension of the drainage by rivers discharging into River Rhine.*

*The applicability of the model is affected by data availability as well. We have access to farily detailed river-profile data facilitating the set up the HEC-RAS model. The emphasis on processes for the sediment generation also matters. Our model assumes simplified sediment generation in agricultural areas due to its small contribution. If a user was more interested in sediment generation on different types of crops, the corresponding processes could be modified.*

*Upon the revision, we will explain the peculiarities of the studied system somewhat better. We will also provide the entire code needed to set up the model as supporting information so that interested readers can adapt and use the code for their own purposes.*

5. The HEC–RAS simulates hourly quasi-steady flow (See LN-15 P-4). What was the reason for not selecting the unsteady option in HEC-RAS? The details about the boundary conditions (e.g., Upstream/downstream) and initial states are missing in the manuscript. What types of boundary conditions you had in your model?
*Response: The temporal resolution of the hydrological model and of HEC-RAS is one hour, because we have hourly gauging and meteorological data. We performed unsteady flow simulations with HEC-RAS, solving the hydrodynamic-wave form of the St.-Venant equations, and did not observe big differences. This may also be attributed to the comparably small size of the catchment so that in-*

*stream retention has only a minor impact on the flow behavior. The unsteady simulations are also less stable. The key outcome of the quasi-steady flow simulations by HEC-RAS is to obtain bottom shear stresses and water depth needed for the modelling of sediment transport in the river channel. The upstream boundary condition was set to time-series of flow and the downstream was set to normal depth. We will add relevant information in the revised manuscript and provide all HEC-RAS files as supplementary material in the revision.*

6. The distances between "computed" cross-sections range from 10 m to 100 m depending on the changes of river bathymetry (See LN-19 P-4). Are these the interpolated cross-section data. What was the interpolation algorithm? Did you use one of in-built (HEC-RAS) interpolation algorithms? Where did you have your observed crosssection data in your model? The details are missing in the manuscript. HEC–RAS model computes the hourly hydraulics for the all cross-sections of the main channel and two major tributaries of the Ammer River (See LN-18 P-4). Does this statement fit the section (i.e., model setup)?
*Response: We have 258 measured cross sections. We have used the built-in interpolation algorithm in HEC-RAS to obtain the additional cross sections (based on linear interpolation between cross-sections), which results in totally 385 cross sections for the entire river network. We will add relevant information in the revised manuscript and provide the bathymetry file in the supplementary material S1.*

7. The section 2.3 needs to be more detailed. Many details are missing in the manuscript. The modeled river schematization needs to be included in the manuscript. How did you include the tributaries in HEC-RAS? The details on the junctions/ confluences are also needed.
*Response: Thanks for this comment. We use HEC-RAS to obtain river hydraulics, so we didn't write too much details about HEC-RAS model considering the article length. The confluence points of all smaller tributaries are points at*

*which the river discharge for the HEC-RAS model changes. For the few conflu-
ences of HEC-RAS modeled streams, we use the standard framework provided
by HEC-RAS. We will provide all details of the HEC-RAS model in the supplemen-
tary material, and add some specifics in the main text of the revised manuscript.*

8. The land use is classified into urban areas and non-urban areas (See LN-23 P-
4). Impervious surfaces such as roads and roofs are regarded as urban areas,
while non-urban areas consist of pervious surfaces such as gardens and "parks",
agricultural areas, and forests (See LN-24 P-4). Does this statement contradict
with your section 3.1(See Table 1)? Did you classify your LULC into urban and
non-urban?
*Response: Table 1 shows the land cover of urban area, agriculture, and forest.
It is used in the hydrological model in terms of different parameters for ET and
storage. Then the agricultural area and forest are combined as non-urban areas
to apply non-urban algorithm for sediment generation. Because we use two algo-
rithms for sediment generation, one is for urban area (including particle build-up
and wash-off) and another for non-urban area (surface runoff induced sediment
production).*

9. The sediment-generating model is used to obtain hourly sediments of different
sources from the 14 sub-catchments (See LN-26 P-4). What is meant by different
sources?
*Response: By "different sources" mean urban and non-urban particles. We will
rewrite it to "The sediment-generating model is used to obtain hourly sediments
of urban and non-urban particles from the 14 sub-catchments" in the revised
version.*

10. We use the urban-area algorithm of SWMM, which "performs well on particle
buildup and wash-off for urban land use", to describe sediment generation from
urban areas (See LN-28 P-4). Does this statement need a reference?

*Response: We will add a reference.*

11. The variables in equation (1) need more description to understand the units (i.e., $gm^{-2}$). The definitions of the variables need to include the area. Moreover, the variable $M(t)$ is not found in the equation. Is equation (1) applicable only for urban areas? What is the reason(s)? Does the equation have a variable to show that it is applicable only for urban areas?

    *The unit $gm^{-2}$ means particle mass per unit area, which indicates current particle build-up in a unit area. The area is used afterwards. After knowing the rate of change of particle mass per unit area, we can use this rate, the time interval, and the area to calculate the mass of particle build-up in the corresponding urban area. M(t) is the same as M, but indicating time dependence. We will use M instead of M(t) to make this variable consistent in the revised manuscript. This equation is only applicable for the build-up of particle in urban areas. Because the urban surface has a capacity (maximum build-up, mass per area) of particles, equation (1) leads to the capacity after several days of the dry period (particle accumulation period). But for the non-urban area, the source of particle is soil, so that we assume an "infinite" source from non-urban areas.*

12. In equation (1), what is the value of "k" used in the model. What is the value of "Mmax" used in the model? The maximum buildup depends on the particle production and cleaning frequency, which is obtained through calibration (See LN-6 P-5). This statement needs more explanation.

    *Response: The values of "k" and "Mmax" used in our model were provided in Table 2 of the manuscript. The maximum build-up varies with cities, which affected by the particle production (such as traffic density, population density, and industry density) and cleaning frequency which takes some urban particle out of the system. Therefore it is a calibration parameter.*

13. In equation (5), what is the value of your critical stress? Does the value of critical

stress vary with time? Don't you consider the particle accumulation in non-urban areas? Is equation (4) applicable for all non-urban areas? Moreover, sin(theta) is not the mean slope of the sub-catchment. Since theta is very small, you will end up saying sin(theta)=tan(theta)?

*Response: The value of critical stress was provided in Table 2 of the manuscript. Yes, the critical stress could vary with time, which is affected by many factors such as the vegetation. But in our case, due to the limited contribution from non-urban areas, we simplified the sediment production from non-urban areas and used a time-independent critical stress. We don't have particle accumulation in non-urban areas, but we assume a "infinite" particle supply from non-urban areas. We would say that for the sake of simplicity the equation (4) can be used for all non-urban area to estimate shear stress. But if the users are focusing on more precise calculation of shear stress on non-urban surfaces, they should search for more precise methods. Yes, when theta is very small, $\sin(\theta) = \tan(\theta)$. We will revise the equation (4) to use $\tan(\theta)$ instead of $\sin(\theta)$.*

14. Is equation (11) formulated by the authors? Otherwise, the reference is required. In equation (11), what is the unit of bank erosion rate? This unit needs to be compared with the unit of bed erosion rate (i.e. equation (10)). Does the bank erosion rate vary spatially along the reach? Does equation (11) cover the bank erosion in the freeboard region? In equation (11), what is the value of your critical shear stress for bank erosion? Does the value of critical stress vary with time? In equation (11), what is the equation of your bank shear stress?

*Response: Yes, equation (11) was formulated based on bank erosion due to excess shear stress. The unit of the bank erosion rate is $gm^{-1}s^{-1}$, which is the same as the unit of the bed erosion rate. Yes, the bank erosion rates vary spatially and temporally along the reach. Equation (11) is the average bank erosion for a cross section, we don't have separate erosion algorithms for freeboard regions, which needs more detailed information on the cross sections, such as vegetation*

*types of freeboard regions. The critical shear stress is provided in Table 3 of the manuscript. The critical stress in our model doesn't vary with time and the shear stress is obtained through the HEC-RAS model.*

15. In equation (10), the units of bed erosion rate and the specific rates of particle and mass erosion are not understood. What is meant by "specific rate"? How do you compare this (i.e., specific rate) with the bed erosion rate? What are the values of the thresholds (i.e., mass erosion threshold and particle erosion threshold)? Does the bed erosion rate vary spatially along the reach?

    *Response: In equation (10), the units of bed erosion and specific rates of erosion are $gms^{-1}$, which means how much mass of particles can be eroded per unit length (1 m) of the river channel per second. The "specific rate" is a constant. If we know the shear stress and the critical shear stress, we can calculate the excess of shear stress, multiply it with the specific rate constant, and obtain the bed erosion rate. The values of the thresholds are provided in Table 3 of the manuscript. Yes, the bed erosion rates vary spatially and temporally along the reach.*

16. Does equation (9) represent the "bank" and "bed" deposition rates? What is the reason to condition based on the bottom shear stress of the river?

    *Response: Equation (9) represents the deposition rate of suspended sediment from the aqueous phase to the bed sediment. When the shear stress generated by the flow is smaller than a critical shear stress, it cannot maintain all sediments in the water phase in suspension, therefore some of the suspended sediments will deposit on the river bed.*

17. In equation (8), what is the assumption(s) made in formulating the first component of the right-hand-side of the equation?

    *Response: The first component of the right-hand-side of equation (8) is the deposition of suspended sediments onto the river bed, which increases the bed*

*sediment mass. The assumption is similar as the response of the question 16).*

18. Are your computations cells of equal size? As per LN-5 in P-6, the computation cells are formed by two cross-sections. However, your cross-sections are not equally spaced (See LN-19 P-4). Wont this influence your model outcome?
*Response: Our computation cells are not equal in size. The computation cells are small in the river segments with rapidly changing bathymetry, while they are big in the river reaches with relatively stable bathymetry (maximum 100 m). The reasons why we don't use equally spaced cells are that 1) if we use cells of equal size, then the minimum spacing (10 m in our case) should be used, otherwise, we cannot well represent the river segments with fast changing geometry by using larger spacing. It makes the number of computation cells almost 10 times bigger, which results in the increase of one order of magnitude of computation time; 2) Using a bigger spacing for the river segment with stable bathymetry is feasible, because the flow characteristics are similar.*

19. As per the title of the manuscript, the catchment is dominated by groundwater. However, the current version of the manuscript does not lead to understand this statement. Does the equations account in your suspended sediment transport model account for this statement (i.e., dominantly groundwater-fed catchment)?
*Response: The water flux of the Ammer catchment is dominated by groundwater inputs (see the stable base flow contrasting other catchments in the area), whereas the sediment load is dominated by urban particles. The dominance of groundwater (plus the sewage treatment plant) on the hydrology is reflected in small surface-runoff contributions to the water flux, which is restricted to only a few events. The latter is the main reason why so little sediments generated in the agricultural areas. We will provide a flow duration curve in the supplementary material to demonstrate the dominantly groundwater-fed property. And we contemplate on modifying the title of the manuscript.*

20. The equation (7) needs to be derived from first principles. Does this equation account for sink (i.e., flow diversion)? Is this equation formulated correctly? Considering your equation (10) and equation (11), what is the unit of the third component in equation (7)? Did you use the equations (1-6) in your equation (7)? Which component of your equation (7) accounts for your equations (1-6)?

*Response: Equation (7) is formulated for the main channel, where no flow diversion exists. In our case, tributaries enter into the main channel, which are regarded as lateral flow (the source term, the last component in the right-hand-side of equation (7)). The unit of the third component in equation (7) is $gs^{-1}$, which is in consistent with the unit of the change rate of suspended sediment mass. We compute the change rate of mass instead of concentration due to numerical reasons. Equation (7) does not explicitly use equations (1-6), but implicitly considers them. Equations (1-6) are used to calculate sediment from the catchment, which is the source term for the sediment transport in the river channel ($c_{lat}^i$ and $Q_{lat}^i$ in equation (7)).*

**References**

Clark, L. A., and Wynn, T. M.: Methods for determining streambank critical shear stress and soil erodibility: Implications for erosion rate predictions, Transactions of the ASABE, 50, 95-106, 2007.

Lammers, R. W., and Bledsoe, B. P.: Parsimonious sediment transport equations based on Bagnold's stream power approach, Earth Surf Proc Land, 43, 242-258, 10.1002/esp.4237, 2018.

Schneider, J. M., Turowski, J. M., Rickenmann, D., Hegglin, R., Arrigo, S., Mao, L., and Kirchner, J. W.: Scaling relationships between bed load volumes, transport distances, and stream power in steep mountain channels, Journal of Geophysical Re-

search: Earth Surface, 119, 533-549, 10.1002/2013jf002874, 2014.

Winterwerp, J. C., van Kesteren, W. G. M., van Prooijen, B., and Jacobs, W.: A conceptual framework for shear flowinduced erosion of soft cohesive sediment beds, Journal of Geophysical Research: Oceans, 117, 10.1029/2012jc008072, 2012.
* * *

---

## Author Comment (AC2) · 8 Mar 2018

*We thank the anonymous reviewer for his/her critical comments helping us to focus on explanations that might be missing.*
*In the following, the reviewer comments are in normal fonts, whereas our responses are in italic.*

[Figure]

**Comments and Responses**

The manuscript illustrates a coupled catchment-stream model for water quantity and sediment productions and transport developed for the Ammer River Basin (Germany), which has some important karstic contribution to baseline flow and suspended sediments. The physics- based model that is proposed includes a complex one dimensional hydraulic component for calculating shear stress. Erosion rates are then based on shear stress concepts applied to erosion of bed and bank material (either deposited sediments or consolidate beds), as well as in-stream deposition. The model was developed to tackle Ammer Basin hydrology in particular, however it is proposed as an integrated model of general applicability. The model is built on components of other hydrological and sediment models. It appears to be very focused on in-stream processes. Conversely, sediment sources from land processes (soil erosion) seems too simplified. I have some concerns about the paper and its content.

1. First of all, it is not true that existing models do not account for in-stream processes (P2 L24). For example, I know that SWAT model offers several ways to tackle in-stream sediment transport and erosion, including some physics based approaches based on shear stress and the possibility to include cross section of reaches. Some literature has shown that these SWAT approaches work well. The authors should therefore revise their statements.
   *Response: The reviewer is right that SWAT has a sediment routine in stream channels for sediment transport. What we meant in the manuscript (P2 L24) is that the models simplify the in-stream processes, such as simplifying the shape of cross sections. HEC-RAS solves the full 1-D St. Venant equation for any type of cross-section including cases with changes in the flow regime. We will sharpen the statement in the revised manuscript.*

2. Sediments from urban land is modelled with a well-known wash off/build up approach. Instead erosion from non-urban areas is tackled with an (to my point of view overly) simplified approach (eqs. 4-6) whose main drivers are runoff and slope. Only one shear stress threshold is considered despite agricultural land diversity, which includes a variety of crops like cereals, vegetables and natural vegetation. This approach does not consider any variability in soil erodibility, or changes in crop cover during the year, which instead impact soil erosion from agricultural land especially among seasons. This flaw limits very much results drawn from the model especially in terms of seasonality and 'hot moments' of erosion.

*The soil erosion varies in different croplands. In our studied catchment, the agricultural area is dominated by non-irrigated arable land (80.2 % of the total agricultural area), the crop of which is mainly cereals (largely corn, but the farmers perform crop rotation). We used a simple approach to estimate the average sediment production from agricultural areas without differentiating the crop types. We have the following reasons:*

*(a) The Ammer catchment is dominantly groundwater-fed. The stream discharge is too small for the width of the valley due to rerouting of the former head-water catchment in the early Pleistocene. Surface runoff from agricultural areas occurs not very often and is weak, which makes the contribution of agricultural sediment generated by surface runoff rather small compared with contribution of the urban particles. This justifies the simplified approach used for the agricultural areas and a well-known approach for urban areas. The dominance of urban particles in the catchment has been reported by Rügner et al. (2013; 2014) and other studies performed by the same workgroup who found a surprisingly high load of persistent organic pollutants in River Ammer, which could be interpreted by the high contribution of urban particles to the total sediment load.*

*(b) To model the variability of soil erosion from different croplands and crop rotation, more information and model parameters are needed. Calibrating the additional parameter sets for different types of croplands would increase the model complexity in an unfeasible manner. We used the same parameterization to estimate the average sediment generation.*

*(c) The crop on the agricultural areas changes from year to year by crop rotation. But the information on the crop rotation is unknown for us, which limits the application of different parameterizations.*

3. I have some concerns about the calibration and validation of the model. The model has 14 calibration parameters and is calibrated vs 1 single station at the outlet of the Basin. I also note that calibrated parameter Ch (Table 2) which regulates the non-urban sediment loads, is lower than its initial range. The risk of over parameterization of this model is very high. Some sensitivity analysis should be shown and discussed as this represent a limit of potential conclusions of the paper. Calibration was driven by a LHS scheme but conducted manually. The authors state that calibration parameter sets were retained to derive 90 % confidence intervals. However these confidence intervals are not shown nor further commented expect for a vague comment at P 11 L 14. The model runs at hourly time step. At what time step calibration and validation were conducted? Water discharge was calibrated for 2013- 2014 and validated for 2015-2016. Sediments were calibrated for 2014 and validated for 2016. Why data for 2015 was not used in this exercise? Data was available as shown in fig 5 but model simulations are not shown. However, model simulations are used for sediment balance considerations e.g. figures 6 and 7. Please explain. The model missed simulation of 2 large rainfall events (one in 2014 and one in 2016), where the highest sediment concentrations occurred. The explanation offered (P11 L 14, p12 L1-2) is that rainfall precipitation measurements 'may be missing'. This should be verified in

the input data. In any case, these 2 events were the most important for sediment load, so all sediment balance is flaw as it cannot consider these main events. It would also be good to see some events in more details given the high temporal discretization of the model.

*We used the LHS scheme to calibrate the hydrological model because it runs very fast. But the sediment transport model was calibrated manually due to the high computation time. Therefore, 90 % confidence intervals were calculated for the hydrological model and are shown in Fig. 4 of the manuscript. The models were calibrated and validated to the daily data. The reason for not using data of 2015 for the sediment transport model is that we don't have measurements in 2015. In Fig. 5 of the manuscript, the red dashed line represents measurements and blank solid line indicates model results. The red dash line shows a data gap for year 2015. The model results of 2014 and 2016 were used to compute the sediment balance. Even though the peak suspended sediment concentrations were not well reproduced by the sediment transport model, the model predicted high suspended sediment concentrations for these two events. The attached figure shows three different events with the peak suspended sediment concentration ranging from 200 mg L$^{-1}$ to 1300 mg $^{-1}$. It demonstrates that the sediment transport model in the manuscript has the capability to predict suspended sediment concentrations for different size of events. It is affected very much by the input data such as precipitation, which drives the surface runoff. We will add the figure to the revised manuscript.*

4. The results of the model indicate that urban land is the major source of sediments in the catchment. this is possible, but I find hard to believe that 67 % of the basin (agricultural land) contributes almost nothing to sediment loads. Even if runoff production is very low and land is gentle sloping (P 21 Lines 10-12), I would expect more contribution. The authors should check with other lines of evidence (e.g. literature of soil erosion from agricultural land in the region) if their results

are realistic.

*Response: In the Ammer catchment, the contribution of sediments from agricultural land is very small for the investigation period (2014-2016). There are several reasons:*

(a) *The formation of the Ammer catchment results in a very wide valley and a small river (the Ammer River) due to rerouting of the former head-water catchment in the Pleistocene. The catchment has a large water storage capacity due to the karst and the slope of this catchment is mild. These characteristics make the surface runoff from the agricultural areas very small, thus small sediment production.*

(b) *The infiltration rates of loamy soil and clay loamy soil are 10–20 mm h-1 and 5–10 mm h-1, respectively. But for the Ammer catchment, very few events have precipitation intensity greater than 10 mm h-1 during the simulation period of 2014–2016. Thus, very few and small surface runoff can occur in the agriculture land, which limits sediment generation and transport from agricultural areas to the river channel.*

(c) *We will add the flow duration curve in the supplementary material. It can be seen that only 0.04 per cent of the discharge is greater than 12 m3 s-1 (2-year return period level), totally 3 events for the entire simulation period, which indicates a very small proportion of big events. During big events, it is possible to generate surface runoff in the agricultural areas.*

(d) *The previous study of Rügner et al. (2013) has compared the turbidity measurements of the same event (30. Nov. 2012) for the Ammer catchment and the Steinlach catchment. The two catchments are in the same region and with similar size of catchment area. The population density of the Ammer catchment is*

*higher than that of the Steinlach catchment (but in the same order of magnitude). The difference lies in the topography of the catchments. The measurements of that event show that the peak turbidity of the Steinlach River is 7.4 times higher than that of the Ammer River. It indirectly indicates that the sediment generation of agriculture land from the Ammer catchment is much less.*

Fig 8 indicates an increase of sediment sources following a power-law with discharge, which may make sense. However, I wonder if an excess of sediment transport capacity of the stream was considered in the model. This may regulate deposition when sediment sources are very high. I do not see this being considered in the model (but I may be wrong). Please discuss.
*Response: The sediment transport capacity is important for bed load transport. For a given discharge, the flow can only transport a limited bed load by rolling, sliding, and hopping, which is regulated by the transport capacity. The bed load material is mainly sand and gravel. But the cohesive sediment transport is different, which is out of the range of applicability of sediment transport functions formulated based on bed load. The transport capacity of cohesive sediments always exceeds supply (Brunner, 2016). The suspended sediment transport of this study belongs to cohesive sediment transport. Therefore, we stole the algorithm for cohesive sediment in HEC-RAS to simulate suspended sediment transport in our matlab model, which is based on shear stress. The previous study also used shear stress related processes to model suspended sediment transport (Li et al., 2008).*

5. What data was used to set karstic sediment loads?
*Response: The turbidity of ≈3 NTU was observed for the periods without runoff events (base-flow conditions) in the Ammer catchment. The study of Rügner et al. (2013) showed that karst springs in the Ammer catchment contribute to turbidity. Other studies also showed that karst systems can contribute suspended sediments (Bouchaoua et al., 2002; Meus et al., 2013). For the Ammer River,*

*the subsurface flow through the karst system dominates the river flow in periods without rainfall events. Therefore, the turbidity under base-flow conditions is potentially generated by subsurface flow through the karst matrix. We set a constant suspended sediment concentration to the subsurface flow. The subsurface flow is obtained from the catchment hydrological model. Then the karstic sediment load was calculated by subsurface flow rates and constant suspended sediment concentrations.*

6. Section 3.1 should precede model description. The model was built for the Ammer and some important information driving model conceptualization is given in this section, so this should come first. Information about measurement data should be given in this section. Please move P 9 Lines 14-17 and P10 Liens 18-25 to after current P 8 line 14.
   *Response: We will revise accordingly in the revised manuscript.*

7. please change color of Load urb and load bed in figs 6 and 7 to better distinguish them.
   *Response: We will revise accordingly in the revised manuscript.*

8. schematic text at P 18-19 lines 12 onward should be given as a table.
   *Response: We will revise accordingly in the revised manuscript.*

9. reference in the conclusion to events with 2-year return period (P 22 L 2) is surprising. No reference to return period is done before in the manuscript. Given that model failed to simulate two large rainfall events of the region, I find it hard to believe this statement.
   *Response: We will give the reference to the return period of events during the simulation period in the conclusion and also will mention that in the section of data source in the revised manuscript. The model does not well capture the peak concentrations of the two events, but the model gives high concentrations (even though not reach the peak concentration of measurements). We provided several*

*events with details (Fig. 1 in question 3) to show that the model can predict high suspended sediment concentrations.*

**References**

Bouchaoua, L., Manginb, A., and Chauve, P.: Turbidity mechanism of water from a karstic spring: example of the Ain Asserdoune spring (Beni Mellal Atlas, Morocco), Journal of Hydrology, 265, 34–42, 2002.

Brunner, G. W.: HEC-RAS, River Analysis System Hydraulic Reference Manual Version 5.0, Institute for Water Resources Hydrologic Engineering Center, Davis, CA US, 2016.

Li, R., Luo, F., and Zhu, W.: The suspended sediment transport equation and its near-bed sediment flux, Science in China Series E: Technological Sciences, 52, 387-391, 10.1007/s11431-008-0175-9, 2008.

Meus, P., Moureaux, P., Gailliez, S., Flament, J., Delloye, F., and Nix, P.: In situ monitoring of karst springs in Wallonia (southern Belgium), Environmental Earth Sciences, 71, 533-541, 10.1007/s12665-013-2760-x, 2013.

Rügner, H., Schwientek, M., Beckingham, B., Kuch, B., and Grathwohl, P.: Turbidity as a proxy for total suspended solids (TSS) and particle facilitated pollutant transport in catchments, Environmental Earth Sciences, 69, 373-380, 10 10.1007/s12665-013-2307-1, 2013.

Rügner, H., Schwientek, M., Egner, M., and Grathwohl, P.: Monitoring of event-based mobilization of hydrophobic pollutants in rivers: calibration of turbidity as a proxy for particle facilitated transport in field and laboratory, Sci Total Environ, 490, 191-198, 10.1016/j.scitotenv.2014.04.110, 2014.

[Figure]

**Fig. 1.** Measured and modelled suspended sediment concentrations for different events (the event becomes bigger from left to right of the figure)

---

## Referee Comment (RC2) · Anonymous Referee #2 · 28 Mar 2018

The manuscript presents an integrated sediment transport model including hydrological, hydraulic, sediment-generating and an in-stream transport model for the Ammer catchment in Germany. The attempt to assemble a fully integrated model and explain sediment dynamics fits nicely in the recent efforts to support integrated water resources management, the topic is absolutely timely.

I see two significant points that absolutely require improvement.

First is the conceptual explanation and model representation of overland sediment transport, where there is a gap between soil mobilisation (erosion) and reaching the streams. Retention during overland transport and its dependence on vegetation cover

is generally neglected and not included in any form, except if we consider it to be deeply hidden in the parameters of equations 4-6. This formulation makes sediment loads independent from landcover, which partially violates the Critical Source Area principle (not wholly, because slope and flow depth are still there), and makes the model unable to identify the impact of different cultivation patterns. Although the optimal solution would be to change the non-urban sediment-generating submodel to something more appropriate for such a large and diverse catchment, the absolute minimum is to mention this deficiency in the discussion.

The second major issue is the problem of identification and the credibility of model results. TSS concentrations were measured at a single site, the Pfäffingen gauge. All subcatchments and their various landcover classes contributed to this single data series through various transport processes including in-stream retention or resuspension. The identification of the contribution of each class requires specifying contrasting behaviour for all source types a priori, otherwise one cannot decide about the importance of each process from a single aggregate TSS series. Here it was done through the model specification, which one can partially debate, but that's not a principal issue. The problem is that the results, e.g. the importance of processes is finally conditional on the model specification, which is not mentioned here at all. Thus, the identified (and thoroughly analysed) contribution of each source is only true when the model assumptions are correct. This must be explicitly stated in the manuscript, considering that the applied model equations are not obviously the right ones (which is not a real problem, it just reflects the subjective decisions of the authors), and the outcome seems a bit strange (negligible agricultural contribution with 60% arable land). An additional point to this concern is the imperfect fit of the model to the observed data. While the fit is not worse than what one can usually achieve with TSS modelling, the uncertainty is large enough to make identification of different sources practically impossible.

A more objective decomposition of sediment dynamics could have been the analysis of time-dynamics, that is the identification of slow-medium-fast responding components

and their precipitation or discharge trigger thresholds, and binding known mechanisms to them afterwards. In the manuscript the same happened in mathematical sense, but it is now stated with high confidence that a certain response pattern is obviously the effect of a certain process, which is simply not proven by fitting a model to the TSS data. Emphasising the subjectiveness of results is therefore advisable.

A logical follow-up study could aim at repeating the same exercise using multiple TSS time-series from different locations possessing different shares of sediment sources. This would strengthen the basis for attributing certain sediment dynamics to specific sources.

SPECIFIC COMMENTS

P2 L12 and L15: USLE is an empirical model of soil loss on the plot scale, it is not applicable to entire subcatchments, but not because of the lack of in-stream mechanisms. USLE cannot deal with heterogeneities along the transport pathways of soil particles, so it cannot model how much retention will occur during overland transport. So if we don't speak about a homogeneous plot stretching right down to the stream, with the same soil quality, cultivation method, slope, rainfall exposure, USLE will be a bad approximation.

P3 L7: "Towards this ends" sounds strange.

P3 L11-12: Dry weather sediments from a WWTP are much different from "normal" particles due to their different particle size distribution and much higher organic carbon content. Would this spoil the estimation of TSS from turbidity?

Figure 1: Would "Rural" be a better alternative to "Non-urban"?

P4 L1: It would be reasonable to introduce the Ammer first, as the following sections contain a lot of specific information, which cannot be judged without knowing the basic characteristics of the catchment and river.

P4 L7: When the aquifer is a karst, is "groundwater" the right term? Wouldn't "fracturewater" be a better description? Or is this a mix of karst and non-karst?

Equation 1: This must assume that t is always restarted at the beginning of the accumulation phases. If this is the case, please mention.

Equations 4-6: This formulation ignores that (i) overland flow path lengths have a serious influence on sediment delivery [>90% of the mobilised sediment accumulates during overland transport], and (ii) surface roughness, permeability, flow concentration affect yield. How does landcover quality affect sediment transport here? How would buffer zones work in such a model? Given these shortcomings, please comment whether the landcover conditions in the Ammer catchment make these equations usable.

Figure 2: Given that cross-sections were 100 m away from each other, travel time between cross-sections must have been between 50 to 200 seconds. Then the hourly time step means that this model was solved for a series of quasi steady states. How did the dynamics of sediment sources relate to these times? Wouldn't this mean that some dynamics of the rapidly responding urban sources was lost due to improper numerical resolution?

P7 L9: A significant part of TSS and turbidity comes from the wash load, which practically never deposits. So it is a rather significant simplification that all fractions deposit at the same rate. Did this cause problems in the model fits?

P9 L 11-12: It would be logical to mention the peak NSE value besides the threshold.

Table 2: The applied value of Ch (0.001) is out of the specified range (0.003-0.05). Why?

Figure 4: Baseflow is perfect (which is a big achievement in a karstic catchment), but discharge peaks are seldom met. What does this mean for the TSS calculations? Most of TSS likes to travel with discharge peaks.

Figure 5: Please use log scale for TSS, this linear scale isn't very informative, the

reader can't figure out if the model was right or wrong.

P14 L4-5: Urban and karstic dominance in TSS loads would be exceptional with 60% arable land (which - with its barren soil surface in certain months - is generally considered as the most erosion-prone land use class, besides construction sites). Can you find specific reasons for this?

P14 L9: These infiltration rates seem to be a bit high. Design values (for example: https://stormwater.pca.state.mn.us/index.php?title=Design_infiltration_rates) for loam are around 8 mm/hr, for clay loam around 1-5 mm/hr, which would change the runoff picture significantly. Can you bring up a reference in support of these high rates?

Figure 10: Would be better to show the NET rate and slope along the river, because the present legend is confusing. Where can one see "NET EROSION"?

P22 L5-9: This paragraph is speculative. First it should be shown with further measurements that the model was right.

---

## Author Comment (AC3) · 10 Apr 2018

*We thank the anonymous reviewer for his/her comments helping us to improve the manuscript.*

*In the following, the reviewer comments are in normal fonts, and our responses are in italic.*

[Figure]

**Comments and Responses**

The manuscript presents an integrated sediment transport model including hydrological, hydraulic, sediment-generating and an in-stream transport model for the Ammer catchment in Germany. The attempt to assemble a fully integrated model and explain sediment dynamics fits nicely in the recent efforts to support integrated water resources management, the topic is absolutely timely.

I see two significant points that absolutely require improvement.

1. First is the conceptual explanation and model representation of overland sediment transport, where there is a gap between soil mobilisation (erosion) and reaching the streams. Retention during overland transport and its dependence on vegetation cover is generally neglected and not included in any form, except if we consider it to be deeply hidden in the parameters of equations 4-6. This formulation makes sediment loads independent from landcover, which partially violates the Critical Source Area principle (not wholly, because slope and flow depth are still there), and makes the model unable to identify the impact of different cultivation patterns. Although the optimal solution would be to change the non-urban sediment-generating submodel to something more appropriate for such a large and diverse catchment, the absolute minimum is to mention this deficiency in the discussion.

   *Response: Several factors may affect overland sediment transport, such as retention, flow path, and the influence of vegetation covers. In our study, we used a simple approach to estimate the amount of surface-runoff generated sediments that can reach the river channel. This approach is adopted from Piro and Carbone (2014), which has shortcomings for explicitly considering all processes on the hillslope scale, but there are several reasons for choosing the simple approach:*

[Figure]

*(a) The Ammer catchment is dominantly groundwater-fed. Surface runoff from agricultural areas occurs seldom and is weak, which is caused by the formation of the Ammer catchment, a very wide valley with a small river (the Ammer River) due to rerouting of the former head-water catchment in the early Pleistocene. This geomorphological setup makes the contribution of agricultural sediments generated by surface runoff rather small compared with the contribution of urban particles. We don't think that refining the sub-model for rural sediments would have a major impact on the overall behavior.*

*(b) The dominance of urban particles in the catchment has been reported by Rügner et al. (2013, 2014) and other studies performed by the same work-group who found a surprisingly high load of persistent organic pollutants in River Ammer, which demonstrates the high contribution of urban particles to the total sediment load. Consequently, the contribution of agricultural sediments must be small.*

*(c) Non-irrigated arable land (80.2 %) dominates the total agricultural area in the Ammer catchment, the crop of which is mainly cereals (largely corn, but the farmers perform crop rotation). Unfortunately, crop rotation information is unknown to us, which limits the application of introducing detailed parameters for different vegetation covers. Introducing more parameters to explicitly account for all processes on hillslopes will increase the model complexity and complicate the model calibration process in an unfeasible way.*

*(d) The dominant fraction of non-irrigated land makes it feasible to use one parameter set to estimate average sediment yield from the rural areas. The slope and runoff depth can implicitly reflect some dependence on land covers and they make sediment yields time-variable and different for sub-catchments.*

*We will explain why we use this simple approach for overland sediment transport, provide evidences from other studies to support our statements, and discuss the*

*deficiency of this approach in the corresponding paragraphs.*

2. The second major issue is the problem of identification and the credibility of model results. TSS concentrations were measured at a single site, the Pfäffingen gauge. All subcatchments and their various landcover classes contributed to this single data series through various transport processes including in-stream retention or resuspension. The identification of the contribution of each class requires specifying contrasting behaviour for all source types a priori, otherwise one cannot decide about the importance of each process from a single aggregate TSS series. Here it was done through the model specification, which one can partially debate, but that's not a principal issue. The problem is that the results, e.g. the importance of processes is finally conditional on the model specification, which is not mentioned here at all. Thus, the identified (and thoroughly analysed) contribution of each source is only true when the model assumptions are correct. This must be explicitly stated in the manuscript, considering that the applied model equations are not obviously the right ones (which is not a real problem, it just reflects the subjective decisions of the authors), and the outcome seems a bit strange (negligible agricultural contribution with 60 % arable land). An additional point to this concern is the imperfect fit of the model to the observed data. While the fit is not worse than what one can usually achieve with TSS modelling, the uncertainty is large enough to make identification of different sources practically impossible.

A more objective decomposition of sediment dynamics could have been the analysis of time-dynamics, that is the identification of slow-medium-fast responding components and their precipitation or discharge trigger thresholds, and binding known mechanisms to them afterwards. In the manuscript the same happened in mathematical sense, but it is now stated with high confidence that a certain response pattern is obviously the effect of a certain process, which is simply not proven by fitting a model to the TSS data. Emphasising the subjectiveness of
results is therefore advisable.

A logical follow-up study could aim at repeating the same exercise using multiple TSS time-series from different locations possessing different shares of sediment sources. This would strengthen the basis for attributing certain sediment dynamics to specific sources.

*Response: Data availability has a large influence on the verification of most hydrological, sediment transport, and pollutant transport models. Especially time-series of spatially variable data are very rare and few. In our study catchment, only one gauge was installed at the outlet of the catchment, where we can obtain time-series of turbidity (which can be converted to suspended sediment concentrations). Then the time-series data were used as the reference to calibrate and validate the combined sediment contributions from different sources. At the time being, we don't have data for sub-catchments, which makes it difficult to verify the sediment dynamics from different sub-catchments. We have started a coordinated project in the catchment last year, which includes installing more measurement stations, so that future data can be used to validate the assumptions regarding the assumed behavior of at least a few sub-catchments.*

*As suggested in the comment, it is possible and feasible to investigate the importance of different processes given the right knowledge of the catchment (the prior). In the present manuscript, maybe the specification of the model is not sufficient. We will give more detailed model specifications in the revised manuscript to support the assumptions of the model. For example, we will add the formation of the catchment and flow duration curve to indicate why surface runoff occurs seldom in the agricultural areas in order to support the small contribution of agricultural sediments. Secondly, we will search for other studies in the studied regions to support our assumptions and results. Thirdly, we will provide the mass balance calculation of hydrophobic compounds (such as Polycyclic Aromatic Hydrocarbons (PAHs), which is mainly particle-facilitated), which gives evidence for*

*particles from different sources because the contamination levels of particles with different origins are substantially different. Finally, we will make the code open so that everyone can make further calculation and model development.*

**Specific Comments**

1. P2 L12 and L15: USLE is an empirical model of soil loss on the plot scale, it is not applicable to entire subcatchments, but not because of the lack of in-stream mechanisms. USLE cannot deal with heterogeneities along the transport pathways of soil particles, so it cannot model how much retention will occur during overland transport. So if we don't speak about a homogeneous plot stretching right down to the stream, with the same soil quality, cultivation method, slope, rainfall exposure, USLE will be a bad approximation.

   *Response: We will reorganize the statement on USLE to make it more precise and clearer in the revised manuscript.*

2. P3 L7: "Towards this ends" sounds strange.

   *Response: We will revise it to "In this study" in the revised manuscript.*

3. P3 L11-12: Dry weather sediments from a WWTP are much different from "normal" particles due to their different particle size distribution and much higher organic carbon content. Would this spoil the estimation of TSS from turbidity?

   *Response: Particle size and composition can affect turbidity and TSS. The sediment transport model computes suspended sediment concentrations. We set suspended-sediment concentrations of WWTP effluent as an input to the model. The TSS—turbidity relationship is obtained at the river gauge, which has already taken different particles from different sources into account. Moreover, because*

*of the small contribution to total discharge from WWTP and the small suspended-sediment concentrations, the influence of WWTP particles has been smoothed out.*

4. Figure 1: Would "Rural" be a better alternative to "Non-urban"?

   *Response: Thanks for the suggestion. We will use "Rural" instead of "Non-urban" in the revised manuscript.*

5. P4 L1: It would be reasonable to introduce the Ammer first, as the following sections contain a lot of specific information, which cannot be judged without knowing the basic characteristics of the catchment and river.

   *Response: Thanks for the suggestion. We will adjust the corresponding structure: study site first, then model description.*

6. P4 L7: When the aquifer is a karst, is "groundwater" the right term? Wouldn't "fracture-water" be a better description? Or is this a mix of karst and non-karst?

   *Response: It is a mixture of karst and non-karst, therefore, we will keep using "groundwater". And the water in a karst system is of course groundwater, too.*

7. Equation 1: This must assume that it is always restarted at the beginning of the accumulation phases. If this is the case, please mention. Equations 4-6: This formulation ignores that (i) overland flow path lengths have a serious influence on sediment delivery [>90 % of the mobilised sediment accumulates during overland transport], and (ii) surface roughness, permeability, flow concentration affect yield. How does landcover quality affect sediment transport here? How would buffer zones work in such a model? Given these shortcomings, please comment whether the landcover conditions in the Ammer catchment make these equations usable.

   *Response: Yes, it is restarted at the beginning of every accumulation period considering the remaining particles after the flush period. We will clarify details on*

*Equation 1 in the revised manuscript. Equations 4-6 calculate the amount of sediments reaching the river channel. This method and formulation are described in the study of Piro and Carbone (2014). Yes, several factors may affect sediment yield from agricultural land, such as flow path lengths, different land use (reflecting surface roughness), and soil permeability. But in our study, we used a simple method to estimate sediments generated by surface runoff that reach the river channel. Equation 4-6 don't explicitly account for all mentioned processes above, but implicitly consider the dependence of land use and soil property by slope and flow depth in the formulation. Reasons for choosing this approach please refer to the response to Major Comment 1. We will discuss the shortcoming and limitation of this method in the revised manuscript.*

8. Figure 2: Given that cross-sections were 100 m away from each other, travel time between cross-sections must have been between 50 to 200 seconds. Then the hourly time step means that this model was solved for a series of quasi steady states. How did the dynamics of sediment sources relate to these times? Wouldn't this mean that some dynamics of the rapidly responding urban sources was lost due to improper numerical resolution?

*Response: The integrated sediment transport model consists of the catchment-hydrological model, the catchment sediment-generating model, and the river sediment-transport model. The time resolution of the catchment hydrological and the sediment generating models is one hour, because we have precipitation data with one-hour resolution. The river hydraulics adapt comparably quick to changing discharge, so that the quasi-steady state mode of HEC-RAS (neglecting in-stream retention of water) with hourly resolution was considered sufficient to calculates river hydraulics. We use ODE23s to solve sediment transport in the river channel, ODE23s includes an adaptive time-step scheme, which uses small steps for rapid changes. We have set the MaxStep in ODE23s 5 minutes. Actually the river sediment transport model can simulate rapid responses. But due to*

[Figure]

*the hourly input from catchment sediment generating model, the river sediment transport model capture hourly sediment dynamics.*

9. P7 L9: A significant part of TSS and turbidity comes from the wash load, which practically never deposits. So it is a rather significant simplification that all fractions deposit at the same rate. Did this cause problems in the model fits?

*Response: In our model, we simulate particles with average size, the settling velocity of which is the same. But deposition rates are different, which depend on particle size and shear stress. Shear stress varies along the channel and is also affected by flow rates. In the river segments with bottom shear stress greater than a threshold, particles remain in suspension. This approach cannot address the dynamics of different-size particles. If varied particle sizes are considered, additional processes may be needed such as flocculation, which makes the model more complex. That is not the focus of this study.*

10. P9 L 11-12: It would be logical to mention the peak NSE value besides the threshold.

*Response: We will calculate peak NSE value and mention that in the revised manuscript. NSE values used in this study is because that the Ammer River is mainly groudwater-fed. The relatively high baseflow affects sediment deposition during low flow conditions. NSE values can reflect the goodness of fit for both high and low flow.*

11. Table 2: The applied value of Ch (0.001) is out of the specified range (0.003-0.05).Why?

*Response: The specified range (0.003-0.05) is from a literature (Gilley et al., 1993), which can be used as a reference for parameter estimation. When we use the value of 0.001, the model fit shows a much higher NSE value than that using parameter values greater than 0.003. That is the reason why it is a little smaller than the literature value.*

12. Figure 4: Baseflow is perfect (which is a big achievement in a karstic catchment), but discharge peaks are seldom met. What does this mean for the TSS calculations? Most of TSS likes to travel with discharge peaks.

    *Response: Honestly some peaks are missing in the model. Several factors can affect prediction of flow rates, the most important is precipitation. The model reproduced many events, however, some thunderstorms in summer months cannot be well captured due to insufficient representation of precipitation data. This is a very common problem. Baseflow also has a big influence on sediment transport in our study catchment. Because 65 % of discharge is less than the annual mean discharge (1 $m^3s^{-1}$) and only 1.5 % of discharge is greater than 2 $m^3s^{-1}$, which makes the contribution of suspended sediments under low flow conditions a relevant fraction to the total sediment transport (around 25 %).*

13. Figure 5: Please use log scale for TSS, this linear scale isn't very informative, the reader can't figure out if the model was right or wrong.

    *Response: We will present the log plot in the revised manuscript.*

14. P14 L4-5: Urban and karstic dominance in TSS loads would be exceptional with 60 % arable land (which - with its barren soil surface in certain months - is generally considered as the most erosion-prone land use class, besides construction sites). Can you find specific reasons for this?

    *Response: There are several reasons that the contribution of agricultural sediments is small for the simulation period (2014-2016):*

    (a) *The formation of the Ammer catchment results in a very wide valley and a small river (the Ammer River) due to rerouting of the former head-water catchment in the early Pleistocene. The catchment has a large water storage capacity due to the karst and the slope of this catchment is mild. These characteristics make the surface runoff from the agricultural areas very small, which explains the small sediment production.*

(b) *The infiltration rates of loamy soil and clay loamy soil are 10–20 mm $h^{-1}$ and 5–10 mm $h^{-1}$ (http:// www.fao.org/ docrep/ S8684E/ s8684e0a.htm), respectively. But for the Ammer catchment, very few events have precipitation intensity greater than 10 mm $h^{-1}$ during the simulation period of 2014–2016. Thus, very few and small surface runoff can occur in the agriculture land, which limits sediment generation and transport from agricultural areas to the river channel.*

(c) *We will add the flow duration curve in the supplementary material. It can be seen that only 0.04 per cent of the discharge is greater than 12 $m^3\ s^{-1}$ (2-year return period level), totally 3 events for the entire simulation period, which indicates a very small proportion of big events. During big events, it is possible to generate surface runoff in the agricultural areas.*

(d) *The previous study of Rügner et al. (2013) has compared the turbidity measurements of the same event (30. Nov. 2012) for the Ammer catchment and the Steinlach catchment, which is a southerly tributary to River Neckar, the confluence of which is also in Tübingen. The two catchments have a similar size. The population density of the Ammer catchment is higher than that of the Steinlach catchment (but in the same order of magnitude). The difference lies in the topography of the catchments. The measurements of that event show that the peak turbidity of the Steinlach River is 7.4 times higher than that of the Ammer River. This observation indirectly indicates that the sediment generation of agricultural land in the Ammer catchment is much smaller than in the paired Steinlach catchment.*

(e) *Rügner et al. (2013, 2014) and other studies performed by the same workgroup found a surprisingly high load of PAH in River Ammer, which could be interpreted by the high contribution of urban particles to the total sediment load. It confirms the dominance of urban particles in the catchment.*

15. P14 L9: These infiltration rates seem to be a bit high. Design values (for

example: https://stormwater.pca.state.mn.us/index.php?title=Design_infiltration_ rates) for loam are around 8 mm/hr, for clay loam around 1-5 mm/hr, which would change the runoff picture significantly. Can you bring up a reference in support of these high rates?

*Response: We will give the reference in the revised manuscript, which is from FAO website (Food and Agriculture Organization of the United Nations). http: //www.fao.org/docrep/S8684E/s8684e0a.htm. These values are used as a reference to indicate why not too much surface runoff occurs in the agricultural areas.*

16. Figure 10: Would be better to show the NET rate and slope along the river, because the present legend is confusing. Where can one see "NET EROSION"?

*Response: We want to show detailed length profile of deposition and erosion in Fig. 10 so that we can see different reasons for net deposition/erosion for different slopes. The net erosion is too small and sporadic in the simulation period, thus it may be confusing in the present figure. We will find a way to make it clearer in the revised manuscript such as adding a subplot to show the net rate and slope along the river as suggested.*

17. P22 L5-9: This paragraph is speculative. First it should be shown with further measurements that the model was right.

*Response: As responded to the previous questions, we will show more evidences and compare with other studies (Rügner et al., 2013, 2014; Schwientek et al., 2017) in our study region to prove that the model is right in the manuscript. Furthermore, we will make the code open so that other people can test it. After doing so, we hope this paragraph would be reasonable.*

**References**

John E Gilley, WJ Elliot, JM Laflen, and JR Simanton. Critical shear stress and critical flow rates for initiation of rilling. *Journal of Hydrology*, 142(1-4):251–271, 1993.

Patrizia Piro and Marco Carbone. A modelling approach to assessing variations of total suspended solids (tss) mass fluxes during storm events. *Hydrological Processes*, 28(4):2419–2426, 2014.

Hermann Rügner, Marc Schwientek, Barbara Beckingham, Bertram Kuch, and Peter Grathwohl. Turbidity as a proxy for total suspended solids (tss) and particle facilitated pollutant transport in catchments. *Environmental earth sciences*, 69(2):373–380, 2013.

Hermann Rügner, Marc Schwientek, Marius Egner, and Peter Grathwohl. Monitoring of event-based mobilization of hydrophobic pollutants in rivers: calibration of turbidity as a proxy for particle facilitated transport in field and laboratory. *Science of the Total Environment*, 490: 191–198, 2014.

Marc Schwientek, Hermann Rügner, Ulrike Scherer, Michael Rode, and Peter Grathwohl. A parsimonious approach to estimate pah concentrations in river sediments of anthropogenically impacted watersheds. *Science of the Total Environment*, 601:636–645, 2017.

---

## Author Response (AR1)

**HESS 2018-42: Contributions of Catchment and In-Stream Processes to Suspended Sediment Transport in a Dominantly Groundwater-Fed Catchment**
**- Rebuttal Letter -**

Yan Liu, Christiane Zarfl, Nandita B. Basu, Marc Schwientek,
Olaf A. Cirpka

June 21, 2018

*We thank the reviewers and the editor for their comments helping us to improve our manuscript. In the following, comments are in normal fonts, and our point-by-point replies are in italic.*
*Upon revision we have made the following major changes to the manuscript and Supporting Material:*

1. *We performed a local sensitivity analysis of sediment transport regarding the coefficients of sediment generation in urban and rural areas to verify whether the parameter-set of the manuscript is the local optimal. The sensitivity analysis is mentioned in Sect. 3.6 of the revised manuscript. Details are provided in the Supplementary Material (Table S1).*

2. *We provide several pieces of additional information to support the small contribution of rural particles to the total suspended-sediment load and small and sporadic rural surface runoff in the Ammer catchment. The supporting information consists of the source diagnosis of suspended sediments through an end-member-mixing analysis based on sediment-bound Polycyclic Aromatic Hydrocarbons (PAH) with different origins (Table S2 and Equations S1-S7), the soil erosion map (Fig. S1) of our study area from the state geological survey of Baden-Württemberg (`http://maps.lgrb-bw.de/`), the implications of small rural surface runoff and weak connection between soils and streams from the study of Schwientek et al. [2013], and the measured and modelled flow duration curve (Fig. S2). The mentioned figures, tables, and equations above have been included in the Supplementary Material. The corresponding discussion has been added in Sect. 4.2 of the revised manuscript.*

3. *We provide a figure (Fig. S3) on the relationships between measured turbidity and discharge for the summer and winter periods to show that temporal variation of the critical shear stress may not be relevant in our catchment. Together with the information mentioned in point 1, we discuss the reasons why refining the sub-model for sediment generation in rural areas does not pay off. Therefore, we kept the simplified approach for rural areas to model the average rural sediment delivery, but the shortcomings of this approach are discussed in Secs. 3.4.2 and 4.2 of the revised manuscript. We also provide detailed time series of suspended-sediment concentrations for a few events (Fig. S4) in the Supporting Material to demonstrate that our model is capable to predict suspended-sediment concentrations for different size of events.*

4. *We provide the entire files for the HEC-RAS model setup in the Supplementary Material. We also make the sediment transport code open so that interested readers can modify the code for their own purposes.*

5. *We adjusted the manuscript structure by introducing the study area first before describing our sediment transport models.*

6. *We changed Fig. 5 to a semilogarithmic plot. The colors of figures 6, 7 and 9 have been changed to more distinguishable colors. The filled areas of Fig. 10 were changed to lines in order to better show the net deposition and erosion. The schematic text on suspended-sediment sources under different flow conditions in Sect. 4.3 has been replaced by Table 4 of the revised manuscript.*

**1 Reply to the Comments of the Editor**

Thank you for submitting the responses to the three comments regarding your manuscript "Contributions of Catchment and In-Stream Processes to Suspended Sediment Transport in a Dominantly Groundwater-Fed Catchment".

You have addressed most of the comments in a manner such that I suggest to revise the manuscript accordingly. however, there are two overarching aspects – the erosion from agricultural land and the source apportionment - that are not convincingly treated. There are several aspect of these themes that require a more thorough improvement.

I list these aspects below:

1. Rev. 1 and 2 expressed doubts that the small contribution by erosion from arable fields was actually true. In your respective response to Rev. 2 you suggest to search for further studies supporting your findings. I highly recommend to do so and would like to point out that there is a high-resolution erosion risk map for Germany available Auerswald et al. [2009]. Additionally, it might be worth contacting local practitioners to obtain region specific knowledge that is not available in the scientific literature.

   *Response: Thanks for the suggestion on the soil-erosion map, which we now include in the Supplementary Material together with two additional pieces of supporting information. We have modified Sect. 4.2 of the revised manuscript to discuss these issues and added material to the Supplementary Material. Specifically, we provide the following evidence:*

   (a) *We diagnosed the source of suspended sediments based on their content of Polycyclic Aromatic Hydrocarbons (PAHs) by an end-member-mixing analysis. Elevated PAH contents are indicators of urban origin. This analysis implies a small contribution of rural particles in the suspended sediments, which is in good agreement with our model results. Please see the calculation in Table S2 and Equations S1-S7 of the Supplementary Material.*

   (b) *Schwientek et al. [2013] investigated the hydrological drivers of nitrate export dynamics in the Ammer catchment based on one-year measurements. They observed dilution patterns of nitrate in the Ammer catchment under both summer and winter conditions. According to their analysis the Ammer catchment could be described as a two-component system, dominated by a base flow rich in nitrate and a very fast, diluting component from urban areas. Connectivity between soils and the stream network was lacking. This finding is in agreement with a small surface-runoff component from rural areas.*

   (c) *We obtained the soil erosion map of our study area from the state geological survey of Baden-Württemberg (`http://maps.lgrb-bw.de/`), which is provided in Fig. S1 of the Supplementary Material. Sites with very high erosion risks in the state of Baden-Württemberg are found in northern (Kraichgau) and southeastern (Oberschwaben) areas as well as along the western slope of the black forest. By contrast, most of our study area exhibits the lowest level of soil-erosion risk according to this survey. These finding support our model results.*

2. Rev. 1 expressed concerns (point 3) about the low value of the Ch parameter and asked for a sensitivity analysis that would show how robust your findings about the relevance of urban versus rural sediment sources would be. You explain that due to the computational burden (please provide quantitative information), the model uncertainty was only calculated for the hydrological part but you ignore the comment on the sensitivity analysis. I think this request by Rev. 1 is solid and you have to provide some calculations that demonstrate the robustness of your findings.

   This directly links to one major concern raised by Rev. 2, which is the non-identifiability of model parameters and model structure based on the available data (Point 2). This severely limits the possibility to actually infer the sediment sources from your model results. It is possible that the results in section 4.2 simply reflect your prior knowledge because you could tune the model such as to produce what you expected to find. You argue that you will provide further evidence that the model assumptions were plausible. While this is very welcome, it should be complemented by a (local) sensitivity analysis demonstrating how the modeled sediment sources vary (or don't vary) with changing model parameters. With such a sensitivity analysis you respond properly to the comments/request by Rev.1 (see Point 3 there) and Rev. 2 (Points 2).

   *Response: The computation time of one complete model run is 2.5 hours. As suggested, we performed a local sensitivity analysis of the sediment transport model, the description of which is now included in Sect. 3.6 of the revised manuscript. The model contains 4 parameters related to the urban and rural inputs. In the sensitivity analysis, we regarded the calibrated parameter set as the base case, we then changed each parameter by -90%, -70%, -50%, -30%, -10%, +30%, +50%, +100%, +200%, +300% to obtain 40 scenarios plus the base case in total (see Table S1 of the Supplementary Material). By running the model*

*scenarios we obtained the fraction of the rural contribution to the annual suspended sediment load, Nash-Suttcliff coefficient of efficiency for the entire suspended sediment data ($NSE_e$), and the NSE values for high suspended sediment concentrations ($\geq 30\ mgL^{-1}$, $NSE_h$).*

*Table S1 in the Supplementary Material shows the result of the local sensitivity analysis. It can be seen that with changing model parameters, the modelled rural contributions changed, ranging between 0.1% and 12.3%. The corresponding $NSE_e$ and $NSE_h$ also change (-3.29 to 0.50).*

- *By decreasing the parameter $M_{max}$, the rural contribution to the suspended sediments increases. However, by changing $M_{max}$ from the base case, the values of both $NSE_e$ and $NSE_h$ decrease.*

- *As similar response can be observed by adjusting parameter $K_w$.*

- *By increasing the parameter $C_h$, we also increase the rural contribution to the suspended particles. However, the $NSE_e$ decreases to negative values even though $NSE_h$ slightly increases.*

- *Finally, the rural contribution to the suspended sediment increases when decreasing $\tau_c$, but the $NSE_e$ values decrease (with a little increase of $NSE_h$).*

*The local sensitivity analysis shows that the parameter set of the base case is at least locally optimal. In the base case the rural contribution to the annual sediment load is 3.5%, which is similar to the result of the PAH-based sediment-source analysis (see the reply to editor comment 1). The range of $C_h$ in the manuscript is taken from Gilley et al. [1993]. This range is strongly affected by soil properties. We also found the study of [Romero et al., 2007] listing smaller $C_h$-values. Therefore we have adjusted the range of $C_h$ in the revised manuscript.*

*To increase the transparency of the paper, we now provide the HEC-RAS files and the Matlab-code of the sediment transport model in the Supplementary Material so that interested readers can use and potentially modify the code.*

3. Along a very similar line, Rev. 2 commented (point 15) that the infiltration rates were high in his opinion and asked how the results would change upon lowering these values. Although you provide a reference for these infiltration rates you fail to provide the more important answer which concerns how your findings would change upon less effective infiltration. Please provide simulation-based evidence for the robustness of your findings (or its absence!).

*Response: Please see the detailed response to point 15 of Rev. 2 below. Of course, less infiltration at identical precipitation levels would increase surface runoff and thus the risk of soil erosion. A key question in this regard is whether the rain intensity exceeds the infiltration capacity of the soil. Therefore a robust hydrological model is the basis of modeling sediment transport. We carried out an uncertainty analysis of the hydrological model in the manuscript, then calibrated and validated the hydrological model to the measured discharge. The model shows a good fit. We also provided flow duration curves of measurements and model results (Fig. S2 of the Supplementary Material), which indicates that the distributions of high and low flow of the model results are consistent with the measurements. They both imply that our hydrological model is robust. The hydrological model is auto-calibrated and -validated. In essence, if the infiltration was less effective, we would have higher discharge peaks and less base flow; but this is not observed.*

*The results of the hydrological model indeed confirm our prior believes (very little surface runoff on agricultural land). In the manuscript we compared precipitation intensity and infiltration rates of different types of soil in order to find a reason to explain why so little rural surface runoff occurs. Perhaps the statement is not very clear so that the reviewers thought infiltration rates were parameters of our model. However, in the hydrological model, the infiltration rates are model results based on other parameters. We didn't perform simulations in which we deliberately adjusted parameters to obtain less infiltration because the corresponding parameter variations have already been included in our uncertainty analysis of the hydrological model.*

*We have revised the statement on infiltration rates in the manuscript to make the point clearer.*

4. Several times, you defend the model structure for erosion on arable land by your prior knowledge saying that runoff hardly occurs (because of low rainfall intensity compared to the infiltration rates and flat topography) and that therefore urban sources dominate sediment input into the stream network (e.g., responses 4 and 13 to S. Mylevaganam, response 4 to Rev. 1, responses 1 and 2 to Rev. 2). This creates a (potentially) vicious circle because you set up the model structure based on your prior knowledge in such a way that the model prevents proving your wrong.

One example illustrating this issue relates to the comment by S. Mylevaganam about the temporal invariance of the critical shear stress (point 13). You mention in your response that you implemented your simplified approach because of the limited non-urban contributions. However, by doing so, you may actually miss important non-urban fluxes, e.g., during winter when for example cereal fields may

be very prone to erosion even under low intensity rain (e.g., [Prasuhn, 2011]) due to high water saturation. Also the German Environmental Protection Agency points out that low intensity rain may be relevant for triggering erosion (see `https://www.umweltbundesamt.de/themen/boden-landwirtschaft/bodenbelastungen/erosion#textpart-3`). Inspection of Fig. 5 in the manuscripts reveals an event where the model severely underestimates the observed sediment load (winter 2016). This might be potentially such a case where due to the seasonal conditions erosion on arable fields may have been relevant. Hence, a critical shear stress that varied with time might have led to a different result regarding the relevance of different sources for sediment delivery.

*Response: We agree that the critical shear stress may vary with time in some catchments. However, in our catchment, we didn't see relevant effects. The following observations support our assessment:*

- *Figure 1 of this rebuttal letter shows the relationships between measured turbidity and discharge for the summer and winter periods. It can be seen that high-discharge events in winter are smaller than in summer. However, for the same discharge, we don't observe higher turbidity values in winter compared to those in the summer season. The latter reveals that the erosion is similar in winter and summer given the same flow rate. Hence, temporal variation of the critical shear stress may not be relevant in our catchment. Therefore, using a time-independent critical shear stress does not introduce a bias in estimating the average sediment delivery.*

[Figure]

Figure 1: Relationships between measured turbidity and discharge for summer season (Left) and winter season (Right).

- *Low-intensity rain may trigger erosion in some regions of Germany with high erosion risk. But the soil-erosion risk of most of the Ammer catchment is in the lowest range (please see the soil-erosion risk maps of the Ammer catchment and state of Baden-Württemberg in the Supplementary Material, Fig. S1), which means that erosion in low-intensity rainfall events is unlikely in the Ammer catchment.*
- *The small and sporadic rural surface runoff is not only our prior knowledge but also the result of the hydrological model, which is well calibrated by and validated against the discharge measurements. Please see our reply to point 1 on other information supporting the small erosion potential in the agricultural land within the Ammer catchment.*

5. Several times, you defend the model structure with the low contribution from arable fields due to sufficient infiltration capacities of the soil such that no critical runoff would occur. This argument is based on the assumption that overland flow is the only relevant trigger for erosion on arable fields. However, splash erosion (e.g., [Fernández-Raga et al., 2017]) may initiate erosion (and with it overland flow) if the soil structure is not sufficiently stable and rain drops cause surface sealing. Upon surface sealing, infiltration rates may drop substantially causing erosion even if on intact soils the infiltration capacity would be sufficient. Because such aspects are neglected, the chosen model structure cannot prove your prior knowledge/assumptions wrong.

*Response: We have carefully checked the articles on splash erosion. Detachment of soil particles by splash erosion is the first stage of soil erosion. Then the detached soil particles have to be transported by surface runoff to the receiving streams. Splash erosion is affected by many factors such as rain drop size, kinetic energy, rain amount and intensity, and soil properties. It is difficult to measure and validate splash erosion. We also realized that with surface runoff occurring, the splash erosion decreases because surface runoff prevents splash erosion happening. To conclude it, splash erosion may increase the risk of soil erosion at the beginning of the events, which can be partly included by giving a smaller critical shear stress. Finally surface runoff is functioning as carriers for soil particles. The critical shear was obtained through calibration in our study. The simplified approach was chosen to estimate average rural particle delivery by surface runoff, which simulates how many particles can be transported to streams. Given several information to support small rural particle delivery (see point 1), we hope it is fair to use this approach to estimate particles from rural areas even it has some shortcomings. We have discussed this approach in Sect. 4.2 in the revised manuscript.*

6. In this context, it is also peculiar that for the priors of the critical shear stress on fields you refer to a non-published Master thesis Bones [2014] developed in the context understanding scouring around foundations of bridges causing failures. There is no argument not to use such information, however given the large numbers of papers specifically dealing with critical shear stress on crop soils (e.g., as a random selection [Léonard and Richard, 2004]).

   *Response: Bones [2014] listed a full table of critical shear stresses ranging from very erodible soil to very resistant soil. In the revised manuscript, we now cite the published article of Léonard and Richard [2004] to set the range of the critical shear stress.*

7. As a minor comment I'd like to add about your argument that crop rotations – for which you don't have specific information in time - would make it difficult to incorporate more complex agronomic aspects into the model (e.g., Rev. 1, point 2). Given the fact that farm size $(0.2 - 0.4 \ km^2$, I guess for this region) is much lower that the size of your sub-catchments $(1.6 - 10.7 \ km^2$ agricultural land) and on single farms the crop mix is rather stable across years (just single fields are cropped differently) cropping patterns for the scale of interest for your model approach would be rather stable in time.

   *Response: We agree that the cropping pattern in our catchment would be rather stable for the simulation period. After providing additional information (see reply to point 1) to support the rather small contribution of rural particles, we think that using our simplified approach to simulate the average sediment delivery from the catchment is feasible. Introducing more parameters to the model would increase the model complexity and make calibration more difficult. Therefore, we kept the simplified approach for agricultural land but we have discussed the shortcomings and limitations of this approach in Sect. 4.2.*

**2  Reply to the Comments of Reviewer #1**

The manuscript illustrates a coupled catchment-stream model for water quantity and sediment productions and transport developed for the Ammer River Basin (Germany), which has some important karstic contribution to baseline flow and suspended sediments. The physics-based model that is proposed includes a complex one-dimensional hydraulic component for calculating shear stress. Erosion rates are then based on shear stress concepts applied to erosion of bed and bank material (either deposited sediments or consolidate beds), as well as in-stream deposition. The model was developed to tackle Ammer Basin hydrology in particular, however, it is proposed as an integrated model of general applicability. The model is built on components of other hydrological and sediment models. It appears to be very focused on in-stream processes. Conversely, sediment sources from land processes (soil erosion) seems too simplified. I have some concerns about the paper and its content.

1. First of all, it is not true that existing models do not account for in-stream processes (P2 L24). For example, I know that SWAT model offers several ways to tackle in-stream sediment transport and erosion, including some physics based approaches based on shear stress and the possibility to include cross section of reaches. Some literature has shown that these SWAT approaches work well. The authors should therefore revise their statements.

   *Response: The reviewer is right that SWAT has a sediment routine in stream channels for sediment transport. We have given a more precise statement on the SWAT model and also sharpened the advantages of using HEC-RAS for calculating river hydraulics of the Ammer River in Sect. 1 of the revised manuscript.*

2. Sediments from urban land is modelled with a well-known wash off/build up approach. Instead erosion from non-urban areas is tackled with a (to my point of view overly) simplified approach (eqs. 4-6) whose main drivers are runoff and slope. Only one shear stress threshold is considered despite agricultural land diversity, which includes a variety of crops like cereals, vegetables, and natural vegetation. This approach

does not consider any variability in soil erodibility, or changes in crop cover during the year, which instead impact soil erosion from agricultural land especially among seasons. This flaw limits very much results drawn from the model especially in terms of seasonality and 'hot moments' of erosion.

*Response: Many factors could affect sediment delivery from rural areas such as land use, soil properties, and rainfall properties. In our study, we used a simplified approach described by Piro and Carbone [2014] to estimate the average sediment delivery from rural areas to streams. Equation 4-6 partly consider the dependence of land use and soil property by slope and flow depth in the formulation. We have the following reasons for choosing this approach:*

- *The end-member-mixing analysis of sediment-bound PAH, the study of Schwientek et al. [2013], and the soil-erosion maps of the state geological survey reflect the small contribution from rural areas to suspended sediments. This information has been added to the Supplementary Material. The discharge-turbidity pattern doesn't differ between summer and winter (see reply to the editor's comment 4). Hence, we don't think that refining the sub-model for rural sediments would have a major impact on the overall behavior.*

- *As pointed out in the response to the editor, there is clear evidence that rural surface runoff is sporadic and small. This is supported by a good match of the hydrological model component and the measured hydrograph. The explanation for the small surface-runoff component is that the slopes of Ammer catchment are very mild. The river valley is too wide for the current river discharge because the river has lost its former head-water catchment in the early Pleistocene. The present upstream end of the catchment is a karstified limestone plateau of the middle Triassic Muschelkalk formation with some loess cover but little topography. This geomorphological setup makes surface runoff on agricultural land and the associated contribution to suspended sediments rather small. Therefore we doubt that refining the model component related to agricultural sediment generation would pay off.*

*Of course, other catchments behave differently and require thus a different focus of a sediment-generation and -transport model. We discuss the limitations of our method in Sect. 4.2 of the revised manuscript.*

3. I have some concerns about the calibration and validation of the model. The model has 14 calibration parameters and is calibrated vs 1 single station at the outlet of the Basin. I also note that calibrated parameter Ch (Table 2) which regulates the non-urban sediment loads, is lower than its initial range. The risk of over parameterization of this model is very high. Some sensitivity analysis should be shown and discussed as this represent a limit of potential conclusions of the paper. Calibration was driven by a LHS scheme but conducted manually. The authors state that calibration parameter sets were retained to derive 90 % confidence intervals. However these confidence intervals are not shown nor further commented expect for a vague comment at P 11 L 14. The model runs at hourly time step. At what time step calibration and validation were conducted? Water discharge was calibrated for 2013- 2014 and validated for 2015-2016. Sediments were calibrated for 2014 and validated for 2016. Why data for 2015 was not used in this exercise? Data was available as shown in fig 5 but model simulations are not shown. However, model simulations are used for sediment balance considerations e.g. figures 6 and 7. Please explain. The model missed simulation of 2 large rainfall events (one in 2014 and one in 2016), where the highest sediment concentrations occurred. The explanation offered (P11 L 14, p12 L1-2) is that rainfall precipitation measurements 'may be missing'. This should be verified in the input data. In any case, these 2 events were the most important for sediment load, so all sediment balance is flaw as it cannot consider these main events. It would also be good to see some events in more details given the high temporal discretization of the model.

*Response: We used the LHS scheme to calibrate the hydrological model because this model runs very fast. By contrast, the sediment transport model was calibrated manually due to the high computation time (2.5 hours for a single run). 90 % confidence intervals were calculated for the hydrological model and are shown in Fig. 4 of the manuscript. The models were calibrated and validated to the daily data. The reason for not using data of 2015 in the sediment transport model is that we don't have measurements in 2015. In Fig. 5 of the manuscript, the red dashed line represents measurements and blank solid line indicates model results. The red dashed line shows a data gap in 2015. The model results of 2014 and 2016 were used to compute the sediment balance. Even though the peak suspended-sediment concentrations were not well reproduced by the sediment transport model, the model predicted high suspended sediment concentrations for these two events. In the revision we have changed Fig. 5 to a semilogarithmic plot.*
*Fig. 2 below shows details of several events. It demonstrates that the sediment transport model in the manuscript has the capability to predict suspended-sediment concentrations for different size of events. Events are affected very much by the input data such as precipitation, which drives the surface runoff. We have added this figure to the Supplementary Material of the revised manuscript (Fig. S4).*
*As suggested, we performed a local sensitivity analysis of the sediment transport model. Please see our*

[Figure]

Figure 2: Measured and modelled suspended-sediment concentrations for different events (the magnitude of the events increases from the left to right of the figure).

*detailed response to the editor comment 2.*

*The reason why we used a smaller value of $C_h$ than the initial range in the manuscript is that, when we used the value of 0.001, the model showed a much higher NSE value than when using the lower limit of the initial range. The range of $C_h$ in the manuscript was taken from Gilley et al. [1993], which was very much affected by specific soil properties. In the meantime, we found the study of [Romero et al., 2007] showing a smaller $C_h$. We have modified the range of $C_h$ in the revised manuscript accordingly.*

4. The results of the model indicate that urban land is the major source of sediments in the catchment. This is possible, but I find hard to believe that 67 % of the basin (agricultural land) contributes almost nothing to sediment loads. Even if runoff production is very low and land is gentle sloping (P 21 Lines 10-12), I would expect more contribution. The authors should check with other lines of evidence (e.g., literature of soil erosion from agricultural land in the region) if their results are realistic.

   *Response: As detailed in the response to the editor's comments , we have several lines of evidence pointing to small rural contributions to the suspended-sediment load: the elevated PAH concentration in the suspended sediments, the nitrate dynamics, the existing soil-erosion-risk map, and the good agreement of the hydrological model to the measured hydrograph. Above, we have also given explanations for the small rural erosion, to which we want to refer here.*

5. Fig 8 indicates an increase of sediment sources following a power-law with discharge, which may make sense. However, I wonder if an excess of sediment transport capacity of the stream was considered in the model. This may regulate deposition when sediment sources are very high. I do not see this being considered in the model (but I may be wrong). Please discuss.

   *Response: The sediment-transport capacity is important for bed load transport. For a given discharge, the flow can only transport a limited bed load by rolling, sliding, and hopping, which is regulated by the transport capacity. The bed-load material is mainly sand and gravel. The transport of cohesive sediments (fines) is different. The transport capacity of cohesive sediments always exceeds supply [Brunner, 2016]. The transport of suspended sediments of our study relates to the cohesive sediment transport. Therefore, we transferred the algorithm for cohesive sediments of HEC-RAS to our matlab model of simulating suspended sediment transport. This algorithm is based on shear stress. This is in line with previous studies using shear-stress related processes to model suspended sediment transport [e.g., Li et al., 2009].*

6. What data was used to set karstic sediment loads?

*Response: We clarified this in Sect. 2.1 of the revised manuscript. We measured a turbidity of 3 NTU under base-flow conditions in the Ammer River. Rügner et al. [2013] showed that karst springs in the Ammer catchment contribute to turbidity. Other studies also showed that karst systems can contribute suspended sediments [Bouchaou et al., 2002, Meus et al., 2014]. For the Ammer River, the subsurface flow through the karst system dominates the river flow in periods without rainfall events. Therefore, the turbidity under base-flow conditions is potentially generated by subsurface flow through the karst matrix. We set a constant suspended-sediment concentration to the subsurface flow. The subsurface flow is obtained from the catchment hydrological model. Then the karstic sediment load was calculated by subsurface flow rates and constant suspended sediment concentrations.*

7. Section 3.1 should precede model description. The model was built for the Ammer and some important information driving model conceptualization is given in this section, so this should come first. Information about measurement data should be given in this section. Please move P 9 Lines 14-17 and P10 Liens 18-25 to after current P 8 line 14.

   *Response: Thanks for the suggestion. We have revised the manuscript accordingly. We now introduce the study area first and then describe the model.*

8. Please change color of $Load_{urb}$ and $load_{bed}$ in figs 6 and 7 to better distinguish them.

   *Response: We have used distinguishable colors for Figures 6 and 7, and changed the corresponding colors of Fig. 9 as well.*

9. Schematic text at P 18-19 lines 12 onward should be given as a table.

   *Response: We summarized the schematic text in Table 4 of the revised manuscript.*

10. Reference in the conclusion to events with 2-year return period (P 22 L 2) is surprising. No reference to return period is done before in the manuscript. Given that model failed to simulate two large rainfall events of the region, I find it hard to believe this statement.

    *Response: We have checked the flood return period on the state discharge monitoring website (LUBW, http://www.hvz.baden-wuerttemberg.de/) and give the reference in Sect. 2.2 (data source). The 2-year return period was modified to 2-year to 10-year return period in the conclusions. The model does not well capture the peak concentrations of the two events, but the model gives high concentrations (even though it does not reach the peak concentration of measurements). We provided several events with details in the supplementary material (Fig. S4) to show that the model can predict high suspended-sediment concentrations. After providing additional evidences such as flow duration curve (Fig. S2), end-member-mixing calculation based on pollutants (PAHs) on different particles (Table S2 and Equations S1-S7), nitrate dynamics implication [Schwientek et al., 2013], and local soil erosion map (Fig. S1), we hope that the statement is sufficiently supported.*

**3 Reply to the Comments of Reviewer #2**

The manuscript presents an integrated sediment transport model including hydrological, hydraulic, sediment-generating and an in-stream transport model for the Ammer catchment in Germany. The attempt to assemble a fully integrated model and explain sediment dynamics fits nicely in the recent efforts to support integrated water resources management, the topic is absolutely timely.

I see two significant points that absolutely require improvement.

1. First is the conceptual explanation and model representation of overland sediment transport, where there is a gap between soil mobilisation (erosion) and reaching the streams. Retention during overland transport and its dependence on vegetation cover is generally neglected and not included in any form, except if we consider it to be deeply hidden in the parameters of equations 4-6. This formulation makes sediment loads independent from landcover, which partially violates the Critical Source Area principle (not wholly, because slope and flow depth are still there), and makes the model unable to identify the impact of different cultivation patterns. Although the optimal solution would be to change the non-urban sediment-generating submodel to something more appropriate for such a large and diverse catchment, the absolute minimum is to mention this deficiency in the discussion.

   *Response: This remark is similar to the comment 2 of reviewer #1, please see our arguments listed there (and in the response to the editor) why we believe that the rural contribution to the suspended-sediment load is indeed so low that further differentiation does not pay off. Equations 4-6 (adopted from Piro and Carbone [2014]) are a simplified approach to estimate the average sediment delivery from rural areas to*

*streams. This formulation does not explicitly consider all known processes on the hillslope scale, but partly considers the dependence of land use and soil property by slope and flow depth in the formulation. We have discussed the limitations of this method in Sect. 4.2 of the revised manuscript. We definitely agree that other catchments will require more elaborate parameterizations of rural soil erosion.*

2. The second major issue is the problem of identification and the credibility of model results. TSS concentrations were measured at a single site, the Pfäffingen gauge. All subcatchments and their various landcover classes contributed to this single data series through various transport processes including in-stream retention or resuspension. The identification of the contribution of each class requires specifying contrasting behaviour for all source types a priori, otherwise one cannot decide about the importance of each process from a single aggregate TSS series. Here it was done through the model specification, which one can partially debate, but that's not a principal issue. The problem is that the results, e.g., the importance of processes is finally conditional on the model specification, which is not mentioned here at all. Thus, the identified (and thoroughly analysed) contribution of each source is only true when the model assumptions are correct. This must be explicitly stated in the manuscript, considering that the applied model equations are not obviously the right ones (which is not a real problem, it just reflects the subjective decisions of the authors), and the outcome seems a bit strange (negligible agricultural contribution with 60 % arable land). An additional point to this concern is the imperfect fit of the model to the observed data. While the fit is not worse than what one can usually achieve with TSS modelling, the uncertainty is large enough to make identification of different sources practically impossible.

   A more objective decomposition of sediment dynamics could have been the analysis of time-dynamics, that is the identification of slow-medium-fast responding components and their precipitation or discharge trigger thresholds, and binding known mechanisms to them afterwards. In the manuscript the same happened in mathematical sense, but it is now stated with high confidence that a certain response pattern is obviously the effect of a certain process, which is simply not proven by fitting a model to the TSS data. Emphasising the subjectiveness of results is therefore advisable.

   A logical follow-up study could aim at repeating the same exercise using multiple TSS time-series from different locations possessing different shares of sediment sources. This would strengthen the basis for attributing certain sediment dynamics to specific sources.

   *Response: The reviewer points to the important general issue that parametric model uncertainty does not cover the conceptual uncertainty and bias of a model. For a rigorous analysis of conceptual model uncertainty one would need several competing models and a tremendously large and informative data set. In our study catchment, only one gauge was installed at the outlet of the catchment, where we can obtain time-series of turbidity (which can be converted to suspended sediment concentrations). Then the time-series data were used as the reference to calibrate and validate the combined sediment contributions from different sources. At the time being, we don't have data for sub-catchments, which makes it difficult to verify the sediment dynamics from different sub-catchments. We have started a coordinated project in the catchment last year, which includes installing more measurement stations, so that future data can be used to validate the assumptions regarding the assumed behavior of at least a few sub-catchments.*
   *However, our conceptual prior that rural particles play a (surprisingly) low role in the Ammer River does not come out of the blue. The arguments have been given in the responses to preceding comments above. We have extended Sec. 4.2 and the Supplementary Material to present and underline them. The actual motivation of the present study was to develop a mechanistic model of sediment-bound PAH transport in the Ammer River, which shows surprisingly high concentrations of PAH (and PCBs) which can only be explained with a high contribution of urban particles (see Table S2 and Equations S1-S7 and Schwientek et al. [2013]). While we may be able to come up with an alternative model in which more particles come from agricultural land, we would have big difficulties to get the PAH load right without making unreasonable assumptions about PAH contamination of rural particles. While the presentation of the PAH model is reserved for a follow-up paper, we have recognized that the arguments based on the PAH need to be properly stated already in the present manuscript. Please notice also the other pieces of evidence supporting little rural surface runoff and associated soil erosion listed in the reply to editor comment 1.*
   *As for parametric uncertainty, we have followed the suggestion of reviewer #1 and performed a local sensitivity analysis (already discussed above).*

**Specific Comments**

1. P2 L12 and L15: USLE is an empirical model of soil loss on the plot scale, it is not applicable to entire subcatchments, but not because of the lack of in-stream mechanisms. USLE cannot deal with heterogeneities along the transport pathways of soil particles, so it cannot model how much retention will occur during overland transport. So if we don't speak about a homogeneous plot stretching right down to

the stream, with the same soil quality, cultivation method, slope, rainfall exposure, USLE will be a bad approximation.

*Response: We have modified the statement on USLE accordingly in the revised manuscript.*

2. P3 L7: "Towards this ends" sounds strange.

*Response: We revised it to "In this study" in the revised manuscript.*

3. P3 L11-12: Dry weather sediments from a WWTP are much different from "normal" particles due to their different particle size distribution and much higher organic carbon content. Would this spoil the estimation of TSS from turbidity?

*Response: Particle size and composition can affect turbidity and TSS. The sediment transport model computes suspended-sediment concentrations. We set suspended-sediment concentrations of WWTP effluent as an input to the model. The TSS-turbidity relationship is obtained at the river gauge, which has already taken different particles from different sources into account. Moreover, because of the small contribution to total discharge from the WWTP and the small suspended-sediment concentrations, the influence of WWTP particles has been smoothed out.*

4. Figure 1: Would "Rural" be a better alternative to "Non-urban"?

*Response: Thanks for the suggestion. We have used "rural" instead of "non-urban" in the revised manuscript.*

5. P4 L1: It would be reasonable to introduce the Ammer first, as the following sections contain a lot of specific information, which cannot be judged without knowing the basic characteristics of the catchment and river.

*Response: Thanks for the suggestion. We have adjusted the corresponding structure. We now introduce the Ammer catchment first, and then describe the model setup.*

6. P4 L7: When the aquifer is a karst, is "groundwater" the right term? Wouldn't "fracture-water" be a better description? Or is this a mix of karst and non-karst?

*Response: It is a mix of karst and non-karst. Water in a karst aquifer is still called "groundwater" (refer to the Karst Hydrogeology and Geomorphology book [Ford and Williams, 2007] and other hydrogeology textbooks).*

7. Equation 1: This must assume that it is always restarted at the beginning of the accumulation phases. If this is the case, please mention. Equations 4-6: This formulation ignores that (i) overland flow path lengths have a serious influence on sediment delivery [>90 % of the mobilised sediment accumulates during overland transport], and (ii) surface roughness, permeability, flow concentration affect yield. How does landcover quality affect sediment transport here? How would buffer zones work in such a model? Given these shortcomings, please comment whether the landcover conditions in the Ammer catchment make these equations usable.

*Response: Yes, it is restarted at the beginning of every accumulation period considering the remaining particles after the flush period. We have added this explanation to equation 1 of the revised manuscript. Several factors may affect sediment yield from agricultural land, such as flow path lengths, different land use (reflecting surface roughness), and soil permeability. But in our study, we used a simple method (Equations 4-6) to estimate sediment delivery to streams by surface runoff in rural areas. The method and formulation are described in the study of Piro and Carbone [2014]. Equations 4-6 don't explicitly account for all processes mentioned above, but implicitly consider the dependence of land use and soil property by slope and flow depth in the formulation. We have given detailed explanations why we believe that the simplified model is sufficient in our application in the reply to comment 2 of reviewer #1.*

8. Figure 2: Given that cross-sections were 100 m away from each other, travel time between cross-sections must have been between 50 to 200 seconds. Then the hourly time step means that this model was solved for a series of quasi steady states. How did the dynamics of sediment sources relate to these times? Wouldn't this mean that some dynamics of the rapidly responding urban sources was lost due to improper numerical resolution?

*Response: The integrated sediment transport model consists of the catchment-hydrological model, the catchment sediment-generating model, and the river sediment-transport model. The time resolution of the catchment hydrological and the sediment generating models is one hour, because we have precipitation data with one-hour resolution. The river hydraulics adapt comparably quick to changing discharge, so that the quasi-steady state mode of HEC-RAS (neglecting in-stream retention of water) with hourly resolution was considered sufficient to calculate river hydraulics. We use ODE23s/ODE45 to solve the system of*

*diffrential equations arising from spatial discretization of the model for sediment transport in the river channel. In the discretization of advection, we use upwinding. ODE23s/ ODE45 includes an adaptive time-step scheme, which uses small steps when needed; we have set the maximum time-step size to five minutes. While the river sediment transport model can simulate rapid responses, we keep the output only at hourly resolution to be consistent with the output of the flow model.*

9. P7 L9: A significant part of TSS and turbidity comes from the wash load, which practically never deposits. So it is a rather significant simplification that all fractions deposit at the same rate. Did this cause problems in the model fits?

   *Response: In our model, we simulate particles with a single, average size, so that the settling velocity is the same for all particles. This is a simplification. A model with distributions of particle sizes would require additional processes such as flocculation, which would make the model more complex. This beyonds the scope of our study.*

10. P9 L 11-12: It would be logical to mention the peak NSE value besides the threshold.

    *Response: We calculated NSE value for high flows, which are important for sediment transport, and the corresponding values has been added in Sect. 3.6 and 4.1. We also kept NSE values in the revised manuscript because the Ammer River is mainly groundwater-fed. The relatively high baseflow affects sediment deposition during low flow conditions. NSE values can reflect the goodness of fit for both high and low flow.*

11. Table 2: The applied value of Ch (0.001) is out of the specified range (0.003-0.05). Why?

    *Response: In the manuscript, the specified range (0.003-0.05) was taken from Gilley et al. [1993], but the NSE-value was much higher with a parameter value of 0.001. As Romero et al. [2007] showed even smaller values of $C_h$, we have modified the range of $C_h$ in the revised manuscript. The new sensitivity analysis of the sediment-generating model confirmed our parameter choice.*

12. Figure 4: Baseflow is perfect (which is a big achievement in a karstic catchment), but discharge peaks are seldom met. What does this mean for the TSS calculations? Most of TSS likes to travel with discharge peaks.

    *Response: In our case, baseflow and peak flow are both important for the sediment transport in the Ammer catchment. Baseflow has a big influence on sediment transport in our study catchment because 65 % of discharge is less than the annual mean discharge (1 $m^3 s^{-1}$) and only 1.5 % of discharge is greater than 2 $m^3 s^{-1}$, which makes the contribution of suspended sediments under low flow conditions a relevant fraction to the total sediment transport (around 25 %). Honestly some peaks are missing in the model. Several factors can affect prediction of flow rates, the most important is precipitation. While the model reproduced many events, some thunderstorms in summer months have not been captured well because the rain gauge missed the spatially localized precipitation event. This is a very common problem. We did a sensitivity analysis for the catchment hydrological model in the manuscript and obtained the best parameter set for discharge simulation. For the revision, we also calculated the NSE for high flows, which indicates an acceptable fit by using hourly precipitation data.*

13. Figure 5: Please use log scale for TSS, this linear scale isn't very informative, the reader can't figure out if the model was right or wrong.

    *Response: We adopted the suggestion and presented log-scale concentrations of suspended sediments in the revised manuscript.*

14. P14 L4-5: Urban and karstic dominance in TSS loads would be exceptional with 60 % arable land (which - with its barren soil surface in certain months - is generally considered as the most erosion-prone land use class, besides construction sites). Can you find specific reasons for this?

    *Response: We have provided additional information to support the small contribution from arable land in the Supplementary Material and in Sec. 4.2 of the revised manuscript. Please see also the reply to similar comments (editor comment 1 and comment 4 of reviewer #1).*

15. P14 L9: These infiltration rates seem to be a bit high. Design values (for example: `https://stormwater.pca.state.mn.us/index.php?title=Design_infiltration_rates`) for loam are around 8 mm/hr, for clay loam around 1-5 mm/hr, which would change the runoff picture significantly. Can you bring up a reference in support of these high rates?

    *Response: We have given a reference from FAO in the revised manuscript (`http://www.fao.org/docrep/S8684E/s8684e0a.htm`). The dominant soil type in the catchment is a luvisol on loess, the state geological (and pedological) survey maps deep infiltration (to a large extent via karst) as the dominant*

*recharge process with some west-east oriented stripes along individual depressions in which surface runoff is considered possible. A lower infiltration capacity would of course lead to more surface runoff and a higher risk of soil erosion, provided that the precipitation exceeds the infiltration capacity. Perhaps the statement on infiltration rates was not clear in the manuscript. In the hydrological model, the infiltration rates are a model outcome. If there was more surface runoff, we would have higher river-discharge peaks and less base flow. However, the hydrological model has been well calibrated and validated to discharge measurements so that we consider it to be robust. In the revision, we have provided additional information in the Supplementary Material such as flow duration curves (Fig. S2) and the PAH-based analysis of particle sources (Table S2 and Equations S1-S7), which all can support little surface-runoff generated soil erosion (in good agreement with our prior knowledge of this catchment). We have revised the statement on infiltration rates to make it clearer.*

16. Figure 10: Would be better to show the NET rate and slope along the river, because the present legend is confusing. Where can one see "NET EROSION"?

    *Response: We have revised the manuscript as suggested. The dash-dotted lines highlight NET rates along the river in Fig. 10. Since the net erosion is too small and sporadic in the simulation period, the red dash-dotted line (representing net erosion) is very close to the X axis.*

17. P22 L5-9: This paragraph is speculative. First it should be shown with further measurements that the model was right.

    *Response: For the revision, we have searched for additional studies to support our results because the direct measurements of turbidity/suspended sediment concentrations are not available. We provided the flow duration curves (Fig. S3), particle source diagnosis based on end-member-mixing calculation of sediment-bound PAHs (Table S2 and Equations S1-S7), some implications from nitrate dynamics [Schwientek et al., 2013], the state soil erosion map (Fig. S1), and a sensitivity analysis (Table S1) to further support our statements. After doing so, we believe the conclusion is reasonable and fair.*

**4 Response to the Comments of Reviewer #3 (S. Mylevaganam)**

1. The catchment input, bed erosion, and bank erosion increase with an increase in flow rates (See LN-18 P-1). Is this a generic statement? What is meant by catchment input? What is the expected relationship between the erosion and flow rate? What is mentioned in the literature? Is it possible to justify this statement (i.e., LN-18 P-1) using equation (10) and equation (11)?

   *Response: By the catchment input we refer to the sum of sediments from urban areas, rural areas, karst system, and waste-water treatment plants (WWTPs), that is, all sediments that are not generated by in-stream processes (bed and bank erosion). For a river with uniform cross section, we would expect a power-law relationship between the erosion and flow rates. Previous studies have shown that the bed load depends on stream power by a power-law function [Lammers and Bledsoe, 2018, Schneider et al., 2014], in which the stream power is a linear function of the flow rate for a given channel geometry. For the entire river system with non-uniform profiles along the reach, however, the cumulative erosion of the river could follow a different functional relationship on flow rate (in general, we expect that the erosion increases with the increase of flow rates, as the bottom shear stress monotonically depends on the flow rate). Equation (10) shows that the bed erosion rate is a piecewise linear function of excess shear stress if the supply of bed sediments is infinite. Equation (11) indicates that the bank erosion rate increases linearly when the shear stress is greater than the threshold. Shear stress increases with the increase of flow rates. Therefore, the bed and bank erosion increase with an increase in flow rates.*

2. Bed erosion and bank erosion are negligible when flow is smaller than the corresponding thresholds of 1.5 $m^3 s^{-1}$ and 2.5 $m^3 s^{-1}$, respectively (See LN-19 P-1). Is this statement about the rate? Moreover, the threshold values (i.e., 1.5 $m^3 s^{-1}$ and 2.5 $m^3 s^{-1}$) need to be normalized using some of the catchment properties to understand the authors' statement. The threshold value on bank erosion (i.e., 2.5 $m^3 s^{-1}$) is greater than the threshold value on bed erosion (i.e., 1.5 $m^3 s^{-1}$). What is mentioned in the literature?

   *Response: Thanks for the good suggestion. The threshold values of bed and bank erosion have been normalized by the mean discharge, leading to critical values that are 1.5 and 2.5 times the mean discharge, respectively. In the manuscript, we studied the effects of flow rate on the total sediment erosion of the entire river. The result indicates a higher threshold of bank erosion than that of bed erosion, which is expected and reasonable. The bank material is more coherent than bed sediments, thus requiring larger shear stress to induce bank erosion compared with bed erosion, which results in a higher threshold of flow rate for bank erosion. Also the literature shows a smaller critical shear stress for bed erosion [Winterwerp et al., 2012] than for bank erosion [Clark and Wynn, 2007].*

3. As per the authors, USLE and SEDD cannot estimate sediment generation by in-stream processes (See LN-15 P-2). Moreover, as per the authors, although SWAT/HSPF/HEC-RAS can simulate "physical processes", none of these models represent in-stream processes well, specifically in natural rivers (See LN-24 P-2). What are those instream processes? In think, the authors need to explain the way the sediment (e.g., suspended) generation and transport is simulated in some of these models (e.g., SWAT) to understand the pitfalls of the existing models to solve the intended problem(s) in Germany.

*Response: The in-stream processes in the manuscript are the deposition of suspended sediment from the water phase to the river bed, bed erosion, and bank erosion due to excess shear stress. We discuss the shortcomings of empirical and physically based models in the revised manuscript. We have also added a statement on the advantages of using HEC-RAS for river hydraulics.*

4. The catchment-scale hydrological model is based on the HBV model (See LN-1 P- 4). Does this statement need a reference? Moreover, the authors have added a quick recharge component and an urban surface runoff component to explain the special behavior of discharge in the Ammer catchment (See LN-4 P-4). The special behavior of discharge in the Ammer catchment and the reason(s) for including the additional components are not understood. Is the integrated sediment transport model applicable anywhere?

*Response: We have added Lindström et al. [1997] as reference to the HBV model. In Sec. 3.2 we have tried to clarify the explanation for adding additional components. We also provide the entire code needed to set up the model as supporting information so that interested readers can adapt and use the code for their own purposes. The reason for adding a quick recharge component is that in the Ammer catchment, the measured hydrograph demonstrates a rapid increase in base flow in sporadic events. We attribute this peculiar behavior to the hydrological functioning of karst with a deep unsaturated zone (distance to groundwater up to 100m at the upstream end of the catchment). In our conceptual model, we assume water storage in the deep unsaturated zone, which spills over when a threshold value is reached, causing quick groundwater recharge to occur which then leads to a rapid increase of base flow. We have added an urban surface runoff component to obtain a surface runoff depth in urban areas in order to simulate particle wash-off from urban land surface. The integrated sediment model can be applied to other catchments with characteristics similar to the Ammer catchment. In particular, the sediments are mainly contributed by urban areas, surface runoff in the agricultural areas is so weak and sporadic that the erosion in the agricultural area can be simplified, and sediment production is driven by surface runoff rather than wind blow. Besides the karst-affected hydrology mentioned above, the Ammer catchment is special because the valley is too wide for the current stream flow. This stems from a different river system (River Nagold) cannibalizing the Ammer in the early Pleistocene. The valley has a size that fits to a stronger river that has lost a substantial fraction of its stream flow. This is why we observe so little erosion on the (too flat) hillslopes. While this situation is special, it is not unique. There are other rivers in South Germany that have lost their original stream flow in the course of the extension of the drainage by rivers discharging into River Rhine. The applicability of the model is affected by data availability as well. We have access to fairly detailed river-profile data facilitating the setup of the HEC-RAS model. The emphasis on processes for the sediment generation also matters. Our model assumes simplified sediment generation in agricultural areas due to its small contribution. If a user was more interested in sediment generation on different types of crops, the corresponding processes should be modified.*

5. The HEC–RAS simulates hourly quasi-steady flow (See LN-15 P-4). What was the reason for not selecting the unsteady option in HEC-RAS? The details about the boundary conditions (e.g., Upstream/downstream) and initial states are missing in the manuscript. What types of boundary conditions you had in your model?

*Response: We added the boundary conditions of HEC-RAS model in Sect. 3.3 of the revised manuscript. We also provide all of the HEC-RAS files in the supplementary material. The reason for choosing a quasi-steady state mode is described below. The temporal resolution of the hydrological model and of HEC-RAS is one hour, because we have hourly gauging and meteorological data. We performed unsteady flow simulations with HEC-RAS, solving the hydrodynamic wave form of the St.-Venant equations, and did not observe big differences. This may also be attributed to the comparably small size of the catchment so that in-stream retention has only a minor impact on the flow behavior. The unsteady simulations are also less stable. The key outcome of the quasi-steady flow simulations by HEC-RAS is to obtain bottom shear stresses and water depth needed for the modelling of sediment transport in the river channel. The upstream boundary condition was set to time-series of flow and the downstream was set to normal depth.*

6. The distances between "computed" cross-sections range from 10 m to 100 m depending on the changes of river bathymetry (See LN-19 P-4). Are these the interpolated cross-section data. What was the interpolation algorithm? Did you use one of in-built(HEC-RAS) interpolation algorithms? Where did you have your observed cross section data in your model? The details are missing in the manuscript.

HEC–RAS model computes the hourly hydraulics for the all cross-sections of the main channel and two major tributaries of the Ammer River (See LN-18 P-4). Does this statement fit the section (i.e., model setup)?

*Response: We added "We have 258 measured cross sections and we used the built-in interpolation algorithm in HEC-RAS to obtain the additional cross sections, which results in totally 385 cross sections for the entire river network." in Sect. 3.3 of the revised manuscript. The sentence, "HEC–RAS model computes the hourly hydraulics for all of cross-sections of the main channel and two major tributaries of the Ammer River" was deleted. More detailed information can be found in HEC-RAS files in the Supplementary Material.*

7. The section 2.3 needs to be more detailed. Many details are missing in the manuscript. The modeled river schematization needs to be included in the manuscript. How did you include the tributaries in HEC-RAS? The details on the junctions/ confluences are also needed.

   *Response: We provided additional details in Sect. 3.3. All HEC-RAS files are provided in the Supplementary Material. We use HEC-RAS to obtain river hydraulics, so we didn't write too much details about the HEC-RAS model considering the article length. The confluence points of all smaller tributaries are points at which the river discharge for the HEC-RAS model changes. For the few confluences of HEC-RAS modeled streams, we use the standard framework provided by HEC-RAS.*

8. The land use is classified into urban areas and non-urban areas (See LN-23 P-4). Impervious surfaces such as roads and roofs are regarded as urban areas, while non-urban areas consist of pervious surfaces such as gardens and "parks", agricultural areas, and forests (See LN-24 P-4). Does this statement contradict with your section 3.1(See Table 1)? Did you classify your LULC into urban and non-urban?

   *Response: Table 1 shows the land cover of urban area, agriculture, and forest. It is used in the hydrological model in terms of different parameters for ET and storage. Then the agricultural area and forest are combined as rural areas to apply rural algorithm for sediment generation. We use two algorithms for sediment generation, one is for urban areas (including particle build-up and wash-off) and another for rural areas (surface runoff induced sediment production).*

9. The sediment-generating model is used to obtain hourly sediments of different sources from the 14 sub-catchments (See LN-26 P-4). What is meant by different sources?

   *Response: By "different sources" we mean urban and rural particles. We revised it to "The sediment-generating model is used to obtain hourly sediments of urban and rural particles from the 14 sub-catchments".*

10. We use the urban-area algorithm of SWMM, which "performs well on particle buildup and wash-off for urban land use", to describe sediment generation from urban areas (See LN-28 P-4). Does this statement need a reference?

    *Response: We added two references [namely Wicke et al., 2012, Gong et al., 2016] in Sect. 3.4.1.*

11. The variables in equation (1) need more description to understand the units (i.e., g $m^{-2}$). The definitions of the variables need to include the area. Moreover, the variable M(t) is not found in the equation. Is equation (1) applicable only for urban areas? What is the reason(s)? Does the equation have a variable to show that it is applicable only for urban areas?

    *Response: We revised equation (1) to make the variable M consistent by using M instead of M(t). The meaning of build-up, "particle mass per unit area", was also added in the context. The unit g $m^{-2}$ means particle mass per unit area, which indicates current particle build-up in a unit area. The area is used afterwards. After knowing the rate of change of particle mass per unit area, we can use this rate, the time interval, and the area to calculate the mass of particle build-up in the corresponding urban area. M(t) is the same as M, but indicating time dependence, and we have changed M(t) to M. This equation is only applicable for the build-up of particles in urban areas. Because the urban surface has a capacity (maximum build-up, mass per area) of particles, equation (1) leads to the capacity after several days of the dry period (particle accumulation period). But for the rural area, the source of particle is soil, so that we assume an "infinite" source from rural areas.*

12. In equation (1), what is the value of "k" used in the model. What is the value of "$M_{max}$" used in the model? The maximum buildup depends on the particle production and cleaning frequency, which is obtained through calibration (See LN-6 P-5). This statement needs more explanation.

    *Response: The values of "k" and "$M_{max}$" used in our model were provided in Table 2 of the manuscript. For the statement on the maximum build-up, we added the information, "The maximum build-up varies with cities, which affected by the particle production (such as traffic density, population density, and industry density) and cleaning frequency which takes some urban particle out of the system. It is obtained through calibration.", into the corresponding paragraph.*

13. In equation (5), what is the value of your critical stress? Does the value of critical stress vary with time? Don't you consider the particle accumulation in non-urban areas? Is equation (4) applicable for all non-urban areas? Moreover, $sin(\theta)$ is not the mean slope of the sub-catchment. Since theta is very small, you will end up saying $sin(\theta)=tan(\theta)$?

*Response: The value of critical stress is provided in Table 2 of the manuscript. The critical stress may vary with time in some catchments and is affected by many factors such as vegetation. However, it is not relevant in the Ammer catchment. From the discharge-turbidity patterns (Fig. 1 in the reply to editor comment 4), we don't observe higher turbidity values in winter compared to that in the summer season for the same discharge. This finding reveals that the erosion is similar for winter and summer given the same flow rate, thus the temporally variable critical shear stress may not relevant in our studied catchment. Therefore, using time-independent critical shear stress is plausible to estimate the average sediment delivery. For more detailed information please refer to the reply to editor comment 4. We don't have particle accumulation in rural areas, but we assume an "infinite" particle supply from agricultural soils. We would say that for the sake of simplicity the equation 4 can be used for all rural areas to estimate shear stress. But if the users are focusing on more precise calculation of shear stress on rural surfaces, they should search for more precise methods. Yes, when theta is very small, $sin(\theta) = tan(\theta)$. We has revised equation 4 to use $tan(\theta)$ instead of $sin(\theta)$.*

14. Is equation (11) formulated by the authors? Otherwise, the reference is required. In equation (11), what is the unit of bank erosion rate? This unit needs to be compared with the unit of bed erosion rate (i.e. equation (10)). Does the bank erosion rate vary spatially along the reach? Does equation (11) cover the bank erosion in the freeboard region? In equation (11), what is the value of your critical shear stress for bank erosion? Does the value of critical stress vary with time? In equation (11), what is the equation of your bank shear stress?

*Response: Yes, equation (11) was formulated based on bank erosion due to excess shear stress. The unit of the bank erosion rate is $gm^{-1}s^{-1}$, which is the same as the unit of the bed erosion rate. Yes, the bank erosion rates vary spatially and temporally along the reach. Equation (11) is the average bank erosion for a cross section, we don't have separate erosion algorithms for freeboard regions, which needs more detailed information on the cross sections, such as vegetation types of freeboard regions. The critical shear stress is provided in Table 3 of the manuscript. The critical stress in our model doesn't vary with time and the shear stress is obtained through the HEC-RAS model.*

15. In equation (10), the units of bed erosion rate and the specific rates of particle and mass erosion are not understood. What is meant by "specific rate"? How do you compare this (i.e., specific rate) with the bed erosion rate? What are the values of the thresholds (i.e., mass erosion threshold and particle erosion threshold)? Does the bed erosion rate vary spatially along the reach?

*Response: In equation (10), the units of bed erosion and specific rates of erosion are $gm^{-1}s^{-1}$, which means how much mass of particles can be eroded per unit length (1 m) of the river channel per second. The "specific rate constant" is a constant. If we know the shear stress and the critical shear stress, we can calculate the excess of shear stress, multiply it with the specific rate constant, and obtain the bed erosion rate. The values of the thresholds are provided in Table 3 of the manuscript. Yes, the bed erosion rates vary spatially and temporally along the reach.*

16. Does equation (9) represent the "bank" and "bed" deposition rates? What is the reason to condition based on the bottom shear stress of the river?

*Response: Equation (9) represents the deposition rate of suspended sediment from the aqueous phase to the bed sediment. When the shear stress generated by the flow is smaller than the critical shear stress, the river cannot maintain all sediments in the water phase in suspension, therefore some of the suspended sediments will be deposited on the river bed.*

17. In equation (8), what is the assumption(s) made in formulating the first component of the right-hand-side of the equation?

*Response: The first component of the right-hand-side of equation (8) is the deposition of suspended sediments onto the river bed, which increases the bed sediment mass. The assumption is similar as in the response to question 16.*

18. Are your computations cells of equal size? As per LN-5 in P-6, the computation cells are formed by two cross-sections. However, your cross-sections are not equally spaced (See LN-19 P-4). Won't this influence your model outcome?

*Response: Our computation cells are not equal in size. The computation cells are small in the river segments with rapidly changing bathymetry, while they are big in the river reaches with relatively stable*

*bathymetry (maximum 100 m). The reasons why we don't use equally spaced cells are that 1) if we use cells of equal size, then the minimum spacing (10 m in our case) should be used, otherwise, we would run into problems in river segments with fast changing geometry. Using fine cells everywhere would increase the number of computation cells by almost a factor 10, resulting in a similar increase of computation time; 2) Using a bigger spacing for the river segment with stable bathymetry is feasible, because the flow characteristics are similar. We have added the information on cross-section interpolation in Sect. 3.3. We also provided all of HEC-RAS files in the Supplementary Material.*

19. As per the title of the manuscript, the catchment is dominated by groundwater. However, the current version of the manuscript does not lead to understand this statement. Does the equations account in your suspended sediment transport model account for this statement (i.e., dominantly groundwater-fed catchment)?

    *Response: The flow duration curve was provided in the supplementary material (Fig. S3). It indicates that 65 % of discharge is less than the annual mean discharge ($1\ m^3 s^{-1}$) and only 1.5 % of discharge is greater than $2\ m^3 s^{-1}$, demonstrating the dominantly groundwater-fed property. The water flux of the Ammer catchment is dominated by groundwater inputs (see the stable base flow contrasting other catchments in the area), whereas the sediment load is dominated by urban particles. The dominance of groundwater (plus the sewage treatment plant) on the hydrology is reflected in small surface-runoff contributions to the water flux, which is restricted to only a few events. The latter is the main reason why so little sediments generated in the agricultural areas.*

20. The equation (7) needs to be derived from first principles. Does this equation account for sink (i.e., flow diversion)? Is this equation formulated correctly? Considering your equation (10) and equation (11), what is the unit of the third component in equation (7)? Did you use the equations (1-6) in your equation (7)? Which component of your equation (7) accounts for your equations (1-6)?

    *Response: We have revised the description of Equation (7) to make it clear that this equation is used for the main channel, where no flow diversion exists. In our case, tributaries enter into the main channel, which are regarded as lateral flow (the source term, the last component in the right-hand-side of equation (7)). The unit of the third component in equation (7) is $gs^{-1}$, which is consistent with the unit of the change rate of suspended sediment mass. We compute the change rate of mass instead of concentration due to numerical reasons. Equation (7) does not explicitly use equations (1-6), but implicitly considers them. Equations (1-6) are used to calculate sediment from the catchment, which is the source term for the sediment transport in the river channel ($c_{lat}^i$ and $Q_{lat}^i$ in equation (7)).*

[revised manuscript text omitted]

---

## Editor Decision (ED1)

Hess-2018-42

Dear Dr. O. Cirpka,

Thank you for submitting the responses to the three comments regarding your manuscript "**Contributions of Catchment and In-Stream Processes to Suspended Sediment Transport in a Dominantly Groundwater-Fed Catchment**".

You have addressed most of the comments in a manner such that I suggest to revise the manuscript accordingly. However, there are two overarching aspects – the erosion from agricultural land and the source apportionment - that are not convincingly treated. There are several aspect of these themes that require a more thorough improvement. I list these aspects below:

- Rev. 1 and 2 expressed doubts that the small contribution by erosion from arable fields was actually true. In your respective response to Rev. 2 you suggest to search for further studies supporting your findings. I highly recommend to do so and would like to point out that there is a high-resolution erosion risk map for Germany available (Auerswald, Fiener et al. 2009). Additionally, it might be worth contacting local practitioners to obtain region specific knowledge that is not available in the scientific literature.

- Rev. 1 expressed concerns (point 3) about the low value of the $C_h$ parameter and asked for a sensitivity analysis that would show how robust your findings about the relevance of urban versus rural sediment sources would be. You explain that due to the computational burden (please provide quantitative information), the model uncertainty was only calculated for the hydrological part but you ignore the comment on the sensitivity analysis. I think this request by Rev. 1 is solid and you have to provide some calculations that demonstrate the robustness of your findings.
  This directly links to one major concern raised by Rev. 2, which is the non-identifiability of model parameters and model structure based on the available data (Point 2). This severely limits the possibility to actually infer the sediment sources from your model results. It is possible that the results in section 4.2 simply reflect your prior knowledge because you could tune the model such as to produce what you expected to find. You argue that you will provide further evidence that the model assumptions were plausible. While this is very welcome, it should be complemented by a (local) sensitivity analysis demonstrating how the modeled sediment sources vary (or don't vary) with changing model parameters. With such a sensitivity analysis you respond properly to the comments/request by Rev.1 (see Point 3 there) and Rev. 2 (Points 2).

- Along a very similar line, Rev. 2 commented (point 15) that the infiltration rates were high in his opinion and asked how the results would change upon lowering these values. Although you provide a reference for these infiltration rates you fail to provide the more important answer which concerns how your findings would change upon less effective infiltration. Please provide simulation-based evidence for the robustness of your findings (or its absence!).

- Several times, you defend the model structure for erosion on arable land by your prior knowledge saying that runoff hardly occurs (because of low rainfall intensity compared to the infiltration rates and flat topography) and that therefore urban sources dominate sediment input into the stream network (e.g., responses 4 and 13 to S. Mylevaganam, response 4 to Rev. 1, responses 1 and 2 to Rev. 2). This creates a (potentially) vicious circle because you set up the model structure based on your prior knowledge in such a way that the model prevents proving your wrong.

One example illustrating this issue relates to the comment by S. Mylevaganam about the temporal invariance of the critical shear stress (point 13). You mention in your response that you implemented your simplified approach because of the limited non-urban contributions. However, by doing so, you may actually miss important non-urban fluxes, e.g., during winter when for example cereal fields may be very prone to erosion even under low intensity rain (e.g., Prasuhn 2011) due to high water saturation. Also the German Environmental Protection Agency points out that low intensity rain may be relevant for triggering erosion (see https://www.umweltbundesamt.de/themen/boden-landwirtschaft/bodenbelastungen/erosion#textpart-3).
Inspection of Fig. 5 in the manuscripts reveals an event where the model severely underestimates the observed sediment load (winter 2016). This might be potentially such a case where due to the seasonal conditions erosion on arable fields may have been relevant. Hence, a critical shear stress that varied with time might have led to a different result regarding the relevance of different sources for sediment delivery.

- Several times, you defend the model structure with the low contribution from arable fields due to sufficient infiltration capacities of the soil such that no critical runoff would occur. This argument is based on the assumption that overland flow is the only relevant trigger for erosion on arable fields. However, splash erosion (e.g., Fernández-Raga, Palencia et al. 2017) may initiate erosion (and with it overland flow) if the soil structure is not sufficiently stable and rain drops cause surface sealing. Upon surface sealing, infiltration rates may drop substantially causing erosion even if on intact soils the infiltration capacity would be sufficient. Because such aspects are neglected, the chosen model structure cannot prove your prior knowledge/assumptions wrong.

- In this context, it is also peculiar that for the priors of the critical shear stress on fields you refer to a non-published Master thesis (Bones 2014) developed in the context understanding scouring around foundations of bridges causing failures. There is no argument not to use such information, however given the large numbers of papers specifically dealing with critical shear stress on crop soils (e.g., as a random selection Léonard and Richard 2004).

- As a minor comment I'd like to add about your argument that crop rotations – for which you don't have specific information in time - would make it difficult to incorporate more complex agronomic aspects into the model (e.g., Rev. 1, point 2). Given the fact that farm size (0.2 – 0.4 $km^2$, I guess for this region) is much lower that the size of your sub-catchments (1.6 – 10.7 $km^2$ agricultural land) and on single farms the crop mix is rather stable across years (just single fields are cropped differently) cropping patterns for the scale of interest for your model approach would be rather stable in time.

In summary, I suggest that you revise the manuscript according to your reply and by seriously taking into consideration the comments described above.

Sincerely

Dr. Christian Stamm
Editor HESS

**References:**

Auerswald, K., P. Fiener and R. Dikau (2009). "Rates of sheet and rill erosion in Germany — A meta-analysis." Geomorphology **111**: 182–193.

Bones, E. J. (2014). Predicting critical shear stress and soil erodibility classes using soil properties Master thesis, Georgia Institute of Technology.

Fernández-Raga, M., C. Palencia, S. Keesstra, A. Jordán, R. Fraile, M. Angulo-Martínez and A. Cerdà (2017). "Splash erosion: A review with unanswered questions." Earth-Science Reviews **171**: 463-477.

Léonard, J. and G. Richard (2004). "Estimation of runoff critical shear stress for soil erosion from soil shear strength." CATENA **57**(3): 233-249.

Prasuhn, V. (2011). "Soil erosion in the Swiss midlands: Results of a 10-year field survey." Geomorphology **126**(1-2): 32-41.